# ASAT: Adaptive Scoring and Thresholding with Human Feedback for Robust Out-of-Distribution Detection

**Daisuke Yamada** *dyamada2@wisc.edu*
*University of Wisconsin–Madison*

**Harit Vishwakarma** *harit.vishwakarma@stats.ox.ac.uk*
*University of Oxford*

**Ramya Korlakai Vinayak** *ramya@ece.wisc.edu*
*University of Wisconsin–Madison*

**Reviewed on OpenReview:** *https://openreview.net/forum?id=4Kd0VMsL76*

## Abstract

Machine Learning (ML) models are trained on in-distribution (ID) data but often encounter out-of-distribution (OOD) inputs during deployment—posing serious risks in safety-critical domains. Recent works have focused on designing scoring functions to quantify OOD uncertainty, with score thresholds typically set based solely on ID data to achieve a target true positive rate (TPR), since OOD data is limited before deployment. However, these TPR-based thresholds leave false positive rates (FPR) uncontrolled, often resulting in high FPRs where OOD points are misclassified as ID. Moreover, fixed scoring functions and thresholds lack the adaptivity needed to handle newly observed, evolving OOD inputs, leading to sub-optimal performance. To address these challenges, we propose *ASAT*, a human-in-the-loop framework that *safely updates both scoring functions and thresholds on the fly* based on real-world OOD inputs. ASAT maximizes TPR while controlling FPR at all times under stationary conditions, even as the system adapts over time. Under nonstationary conditions, the method adapts to distribution shifts with only transient FPR violations during the adaptation period. We provide theoretical guarantees for FPR control under stationary conditions and present extensive empirical evaluations on OpenOOD benchmarks to demonstrate that our approach outperforms existing methods by achieving higher TPRs while maintaining FPR control.

## 1 Introduction

Machine learning (ML) models are typically trained under the closed-world assumption—test-time data comes from the same distribution as the training data (in-distribution, denoted by ID). However, during deployment, models often encounter out-of-distribution (OOD) inputs, such as data points that do not belong to any training class in classification. In safety-critical domains like medical diagnosis, it is crucial that the system refrains from generating predictions for OOD inputs. Instead, these inputs must be flagged so that expert human intervention is sought. To achieve this, an OOD detector is integrated into systems to monitor real-time samples and flag those that appear OOD, ensuring that unreliable predictions are not produced.

To this end, many OOD detection methods have been developed to distinguish OOD inputs from ID inputs (Yang et al., 2022). While ID data is available during model training, diverse OOD examples remain unseen until deployment. This limited access to OOD data naturally leads to an ID-based detection approach: (1) a scoring function is designed or selected to quantify how likely an input is to be OOD (or ID) *using only ID data*, and (2) a threshold is set on these scores—*again based solely on ID data*—to achieve a desired true positive rate (TPR) (e.g., 95% of ID inputs are correctly identified as ID). While this approach has shown promising results, it suffers from the following issues:

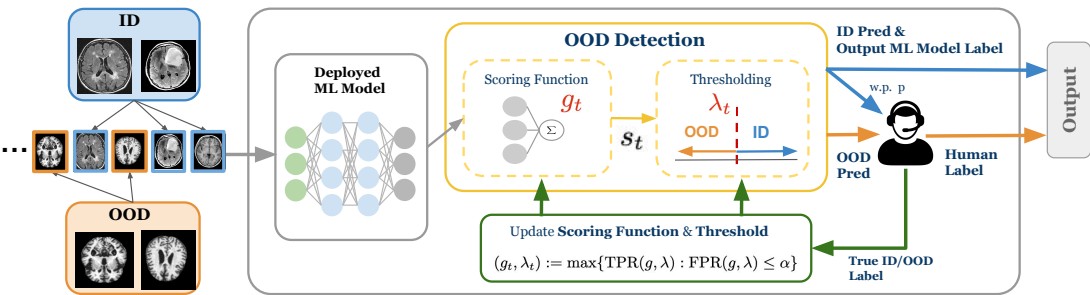

Figure 1: Overview of the ASAT workflow, updating *both* scoring function $g_t$ and threshold $\lambda_t$ via human feedbacks. The ID data consists of brain MRI scans from healthy individuals and those with tumors. The OOD data includes brain scans with other non-tumorous conditions—for example, scans from patients with Alzheimer's disease.

**1) High False Positive Rate (FPR).** OOD detection systems are prone to high FPRs, where OOD inputs are misclassified as ID (Nguyen et al., 2015; Amodei et al., 2016). In high-risk applications, false positives are often more critical than false negatives. For example, detecting non-existent cancer may simply trigger further analysis, whereas missing an actual case can be fatal. Thus, systems in such domains must have strict FPR control (e.g., below 5%). However, the common approach sets score thresholds solely using ID data to meet a target TPR, leaving FPR uncontrolled. As a result, FPRs can be very high—for instance, ranging from 32% to 91% when CIFAR-10 is used as ID in the OpenOOD benchmark (Yang et al., 2022).

**2) Lack of Adaptivity.** During deployment, systems encounter diverse, novel OOD inputs. However, no single scoring function works well for all types of OOD data. As a result, the common approach of relying on a pre-designed scoring function and threshold risks being stuck with one that performs poorly on newly encountered OOD inputs—thereby limiting achievable TPR. These methods miss the opportunity to improve their scoring functions based on OOD inputs observed during deployment (Figure 2). Thus, real-world systems must *adapt their scoring functions and thresholds over time* to handle diverse OOD inputs, maximizing TPR while keeping FPR under control. These challenges motivate the following goal.

> **Goal:** Develop a human-in-the-loop OOD system that strictly controls FPR, *adaptively* maximizes TPR, and thus minimizes human intervention.

**Our Contribution.**   Toward this goal, we make the following contributions:

1. We introduce a framework, ASAT (adaptive scoring, adaptive threshold), that leverages human feedback to identify OOD inputs and *simultaneously update scoring functions and thresholds*, thereby maximizing TPR while ensuring strict FPR control throughout deployment (Figure 1).
2. We provide a theoretical analysis that guarantees FPR control at all times, even as scoring functions and thresholds are updated. Our analysis shows that our approach can maintain FPR below a user-specified tolerance $\alpha$ (e.g., 5%) when the OOD remains constant.
3. We present extensive empirical evaluations under both stationary (OOD fixed) and nonstationary (OOD changing over time) conditions. On OpenOOD benchmarks, we show our framework consistently outperforms existing approaches, achieving higher TPR while ensuring strict FPR control.

We refer to our method as **ASAT** (*adaptive scoring, adaptive threshold*), in contrast to the common approach **FSFT** (*fixed scoring, fixed threshold*), which uses fixed scoring functions and thresholds derived from ID data. We also compare with **FSAT** (*fixed scoring, adaptive threshold*), which updates thresholds while keeping scoring functions fixed. Refer to Section 3.3 for descriptions of FSAT. See Table 1 for a summary.

| Method | Score Fn | Threshold | FPR $\leq \alpha$ | TPR |
|---|---|---|---|---|
| **FSFT** | Fixed | Fixed | ✗ | Fixed (95%) |
| **FSAT** | Fixed | Adaptive | ✓ | Limited |
| **ASAT** (ours) | Adaptive | Adaptive | ✓ | Improved |

Table 1: Comparison of FSFT, FSAT, and ASAT across scores, thresholds, FPR control, and TPR.

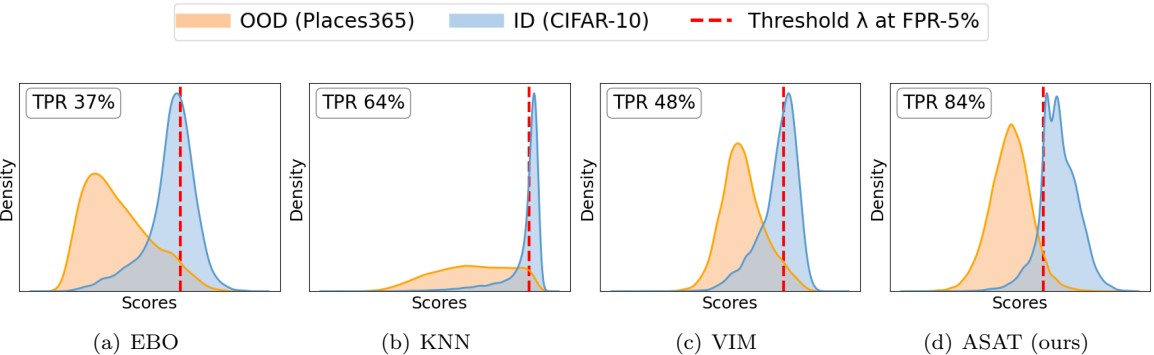

Figure 2: Comparison of score distributions between post-hoc fixed scoring functions (trained solely on ID data) and our learned function $g^*$ from (P2) (trained offline on both ID and OOD). CIFAR-10 and Places365 are the ID and OOD datasets, respectively. As $t$ increases, the ASAT-adapted scoring function achieves higher TPR at 5% FPR compared to EBO, KNN, and VIM methods from the Open-OOD benchmarks. This comparison suggests that incorporating human-labeled OOD inputs encountered during deployment enables ASAT to update scoring functions that better align with real-world OOD points. For more details, see Section 5.

## 2 Preliminaries

### 2.1 Problem Setting

**Data Stream.** Let $\mathcal{X} \subseteq \mathbb{R}^d$ and $\mathcal{Y} = \{0, 1\}$ denote feature and label spaces, respectively, where 0 represents OOD and 1 represents ID. Let $\mathcal{D}_0$ and $\mathcal{D}_1$ be the *unknown* distributions of the OOD and ID data over $\mathcal{X}$. At each time $t$ during deployment, our model sequentially receives a sample $x_t$ drawn independently from the mixture model, $x_t \overset{\text{i.i.d.}}{\sim} \gamma \mathcal{D}_0 + (1 - \gamma)\mathcal{D}_1$, where the mixture rate $\gamma \in (0, 1)$ is fixed but unknown. Let $y_t \in \mathcal{Y}$ be the true label for $x_t$, indicating whether it is an OOD or ID point.

**Scoring Functions.** Let $\mathcal{G} \subset \{g : \mathcal{X} \to \mathcal{S} \subseteq \mathbb{R}\}$ denote a class of scoring functions. For each $x_t$, a scoring function $g \in \mathcal{G}$ computes the score $s_t := g(x_t) \in \mathcal{S}$. We assume that higher scores correspond to a higher likelihood of $x_t$ being ID. Then, the OOD classifier is defined by the threshold function $h_\lambda : \mathcal{S} \to \mathcal{Y}$ as $h_\lambda(s_t) := \mathbb{1}\{s_t > \lambda\}$, with $\lambda \in \Lambda$. The prediction label is $\widehat{y}_t := h_\lambda(s_t)$. The parameter $\lambda$ defines the threshold for classifying $x_t$ as either OOD or ID. Thus, setting $\lambda$ carefully is crucial to ensure safety in OOD detection.

**Population-level FPR and TPR.** Given any $g \in \mathcal{G}$ and $\lambda \in \Lambda$, the *true* FPR and TPR for scores are defined as

$$\begin{aligned} \text{FPR}(g, \lambda) &:= \mathbb{E}_{x \sim \mathcal{D}_0}[\mathbb{1}\{g(x) > \lambda\}], \\ \text{TPR}(g, \lambda) &:= \mathbb{E}_{x \sim \mathcal{D}_1}[\mathbb{1}\{g(x) > \lambda\}]. \end{aligned} \tag{1}$$

For a fixed $g$, both FPR and TPR are monotonically decreasing in the threshold $\lambda$ (Figure 3). Thus, maximizing TPR while maintaining a low FPR naturally involves a trade-off. For fixed $g$, this trade-off is governed by the *overlap of score distributions*; a better $g$ induces more separation between ID and OOD scores. This motivates selecting the pair $(g, \lambda)$ to maximize TPR while keeping FPR below a desired level.

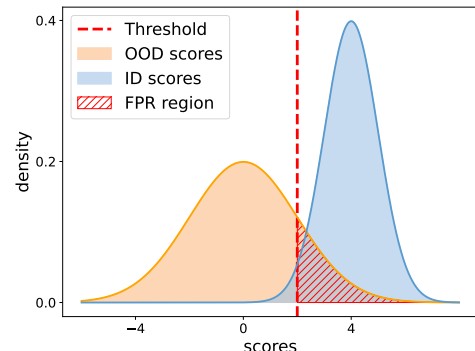

Figure 3: An example of OOD and ID score distributions mapped by fixed $g$, with a decision threshold $\lambda$. FPR corresponds to the hatched region.

In FSFT, both $g$ and $\lambda$ remain fixed for all $t \geq 1$. In FSAT, the scoring function $g$ is fixed a priori and only the threshold $\lambda_t$ is adapted over time to control FPR. In contrast, ASAT aims to update both $g_t$ and $\lambda_t$ using incoming OOD inputs, thereby better controlling FPR and maximizing TPR.

## 3 Methodology

We propose ASAT (Figure 1) that adapts both scoring functions and thresholds using human-labeled OOD samples during deployment. This adaptation is crucial, as the initial scoring function and threshold—typically developed solely on ID data—can yield high FPR or suboptimal performance on certain OOD cases.

### 3.1 ASAT Workflow

At each time $t$, the system processes an input $x_t$ using scoring function $\widehat{g}_{t-1}$ and threshold $\widehat{\lambda}_{t-1}$, both learned at the previous time $t-1$ (see Section 3.2 for the learning process). It first computes score $s_t = \widehat{g}_{t-1}(x_t)$ for $x_t$. If $s_t \leq \widehat{\lambda}_{t-1}$, then $x_t$ is considered OOD and receives a human label $y_t \in \{0, 1\}$. If $s_t > \widehat{\lambda}_{t-1}$, $x_t$ is considered ID, and in this case, we obtain the human label $y_t$ only with probability $p \in (0, 1)$. We refer to this selective sampling as *importance sampling*. This importance sampling enables the detection of OOD shifts and allows for unbiased FPR estimation, which is useful for maintaining FPR control.

### 3.2 Learning Scoring Functions and Thresholds

In this section, we describe how ASAT learns scoring functions and thresholds at each time $t$. We first define the optimal scoring function and threshold based on the TPR and FPR in equation 1, which are unknown in practice. We then present a tractable and practical learning method only using finite samples and estimates.

**Optimal Scoring Function and Threshold.** ASAT aims to maintain strict FPR control during deployment while minimizing human intervention. In practice, accurately predicting IDs reduces the need for human verification. Hence, maximizing TPR effectively minimizes human intervention. With this, we formulate the ASAT objective as a joint optimization problem to find the optimal $(g^\star, \lambda^\star)$ that maximizes TPR under the constraint FPR $\leq \alpha$.

$$g^\star, \lambda^\star := \underset{g \in \mathcal{G}, \lambda \in \Lambda}{\arg\max} \quad \mathrm{TPR}(g, \lambda) \quad \text{s.t.} \quad \mathrm{FPR}(g, \lambda) \leq \alpha. \tag{P1}$$

Here, $\alpha$ is the user-specified FPR tolerance level (e.g., $\alpha = 0.05$ implies FPR must be below 5% at all times). However, since $\mathcal{D}_0$ and $\mathcal{D}_1$ are unknown, the true values of TPR and FPR for a given $(g, \lambda)$ are not accessible, making (P1) intractable. Thus, we estimate FPR and TPR from finite samples collected by time $t$.

**FPR and TPR Estimates.** We assume that ID data is largely available, whereas diverse OOD data is not accessible before deployment. Thus, while the ID data remains a fixed i.i.d. set from training, ASAT incorporates human feedback to identify real-world OOD points during deployment. Let $X_t^{(ood)}$ denote the OOD samples collected up to time $t$ (see Section 3.1), and let $X_0^{(id)}$ be the fixed ID dataset. Then, for a given scoring function and threshold pair $(g, \lambda)$, the FPR estimate at time $t$ is defined as

$$\widehat{\mathrm{FPR}}_t(g, \lambda) := \frac{\sum_{u=1}^t Z_u \cdot \mathbb{1}\{g(x_u) > \lambda\}}{\sum_{u=1}^t Z_u}, \tag{2}$$

where

$$Z_u := \begin{cases} 1 & \text{if } s_u \leq \widehat{\lambda}_{u-1}, i_u = 0, y_u = 0, \\ \frac{1}{p} & \text{if } s_u > \widehat{\lambda}_{u-1}, i_u = 1, y_u = 0, \\ 0 & \text{otherwise,} \end{cases}$$

and $i_u$ indicate whether $x_u$ was importance sampled at time $u$. Also, we estimate TPR using static $X_0^{(id)}$ as

$$\widehat{\mathrm{TPR}}(g, \lambda) := \frac{1}{|X_0^{(id)}|} \sum_{x \in X_0^{(id)}} \mathbb{1}\{g(x) > \lambda\}. \tag{3}$$

Note for a fixed $g$ and $\lambda$, $\widehat{\mathrm{TPR}}(g, \lambda)$ is constant over time, since $X_0^{(id)}$ is fixed.

**Relaxations of (P1).** Using the TPR and FPR estimates from equation 2 and equation 3, we can formulate a tractable optimization problem for the ASAT objective. However, because the indicator sums are non-differentiable and difficult to optimize, we further approximate them using a sigmoid function defined as $\sigma(\kappa, z) := 1/(1 + \exp(-\kappa z))$, $\kappa > 0$, which satisfies $\lim_{\kappa \to \infty} \sigma(\kappa, z) = \mathbb{1}\{z > 0\}$. This yields the following *smooth surrogate* estimates of the TPR and FPR:

$$\widehat{\mathrm{FPR}}_t(g, \lambda) := \frac{\sum_{u=1}^t Z_u \cdot \sigma(\kappa, g(x_u) - \lambda)}{\sum_{u=1}^t Z_u}, \tag{4}$$

$$\widetilde{\mathrm{TPR}}(g, \lambda) := \frac{1}{|X_0^{(id)}|} \sum_{x \in X_0^{(id)}} \sigma(\kappa, g(x) - \lambda). \tag{5}$$

The problem obtained by replacing the true FPR and TPR in (P1) with these surrogates is a non-convex constrained optimization problem. Although this type of problem has been widely studied (Park and Van Hentenryck, 2023; Jia and Grimmer, 2022), it remains challenging to solve efficiently in practice. Therefore, we reformulate it in a regularization form as follows.

$$\widehat{g}_t, \widehat{\lambda}'_t := \arg\min_{g \in \mathcal{G}, \lambda \in \Lambda} -\widetilde{\mathrm{TPR}}(g, \lambda) + \beta \cdot \widehat{\mathrm{FPR}}_t(g, \lambda) \tag{P2}$$

The hyperparameter $\beta \geq 0$ controls the trade-off between maximizing TPR and minimizing FPR. This formulation enables efficient gradient-based optimization of $g$ and $\lambda$. With this, at each time $t$, ASAT estimates TPR and FPR using equation 4 and equation 5, then it solves the problem (P2). However, some issues remain to be addressed.

**Periodically Learning Scoring Functions.** Solving (P2) at every time $t$ is computationally expensive, especially when only a few new OOD points have been identified. To address this, ASAT optimizes (P2) only periodically—when enough new OOD samples have been collected since the last update. Let $U_t$ denote the number of updates to $g$ performed by time $t$, and define $\omega_i^{ood}$ as the user-specified *optimization frequency* for $(i+1)$th update (e.g., $\omega_1^{ood}$ indicates the number of *new* OOD samples required for the second update). Define $\Delta_t^{ood}$ as the number of *new* OOD samples identified by time $t$ *since the last update*. At time $t$, ASAT updates its scoring function only if $\Delta_t^{ood} \geq \omega_{U_t}^{ood}$.

**Model Selection.** Assume an update occurs at time $t$ (i.e., $\Delta_t^{ood} \geq \omega_{U_t}^{ood}$) by solving (P2). However, the newly learned $\widehat{g}_t$ may underperform $\widehat{g}_{t-1}$ in terms of TPR at a given $\lambda$. To prevent this, we do a model selection by applying the TPR criterion:

$$\widehat{\mathrm{TPR}}(\widehat{g}_t, \widehat{\lambda}_t) - 2\zeta(\delta, |X_0^{(id)}|) > \widehat{\mathrm{TPR}}(\widehat{g}_{t-1}, \widehat{\lambda}_{t-1}).$$

Here, $\widehat{\lambda}_t$ is obtained from (Q1) (see Section 3.3), and $\zeta(\delta, |X_0^{(id)}|)$ is the Dvoretzky–Kiefer–Wolfowitz (DKW) confidence interval, valid w.p. at least $1 - \delta$ (Dvoretzky et al., 1956). If this is violated, we let $\widehat{g}_t = \widehat{g}_{t-1}$.

**Threshold Estimation.** The solution $(\widehat{g}_t, \widehat{\lambda}'_t)$ obtained from (P2) may violate the FPR constraint in (P1), since it is unconstrained. That is, it could yield $\mathrm{FPR}(\widehat{g}_t, \widehat{\lambda}'_t) > \alpha$, which undermines our goal of keeping FPR below $\alpha$ at all times in ASAT. To strictly enforce the FPR constraint, we introduce a new step: we estimate a *revised threshold* $\widehat{\lambda}_t$ for $\widehat{g}_t$ using the threshold estimation method proposed by Vishwakarma et al. (2024c).

### 3.3 Learning Thresholds

To ensure that the FPR constraint is satisfied throughout the deployment, we leverage the threshold estimation method FSAT (*fixed scoring, adaptive threshold*). See Table 1 for details.

**FSAT for Threshold Estimation.** The high-level idea of FSAT is to find a safe threshold for a *fixed* scoring function at each time $t$ such that FPR in equation 1 is below $\alpha$. Given $(\widehat{g}_t, \widehat{\lambda}'_t)$ from (P2), we *discard*

the potentially unsafe $\widehat{\lambda}'_t$ and compute a safe threshold $\widehat{\lambda}_t$ via the FPR estimate from equation 2. We achieve this by constructing a uniformly valid upper confidence bound (UCB) on the error between the estimated and true FPR.

**Upper Confidence Bound (UCB).** At each time $t$, the UCB is constructed as

$$\psi_t(\delta) := \sqrt{\frac{3c_t}{N_t^{(o)}} \left[ 2 \log \log \left( \frac{3c_t N_t^{(o)}}{2} \right) + 2 \log \left( \frac{4U_t |\Lambda|}{\delta} \right) \right]}, \tag{6}$$

where $\delta \in (0,1)$ is the failure probability, and $c_t$ is a time-dependent variable. Intuitively, the UCB guarantees that w.p. at least $1 - \delta$,

$$\widehat{\mathrm{FPR}}_t(\widehat{g}_t, \lambda) \in \left[ \mathrm{FPR}(\widehat{g}_t, \lambda) - \psi_t(\delta), \mathrm{FPR}(\widehat{g}_t, \lambda) + \psi_t(\delta) \right],$$

for all times $t \geq 1$ and thresholds $\lambda \in \Lambda$. That is, with high probability, the FPR estimate in equation 2 is within $\psi_t(\delta)$ of the true FPR. In Section 4, we formally prove the FPR guarantee for a calibration-safe variant of ASAT, provide further details, and demonstrate that the UCB is valid for time-varying scoring functions. This leads to the optimization problem for adapting thresholds for a given $\widehat{g}_t$ in ASAT as

$$\widehat{\lambda}_t := \underset{\lambda \in \Lambda}{\arg\min} \quad \lambda \quad \text{s.t.} \quad \widehat{\mathrm{FPR}}_t(\widehat{g}_t, \lambda) + \psi_t(\delta) \leq \alpha \tag{Q1}$$

The problem (Q1) can be directly solved via binary search. Since TPR is monotonically decreasing in $\lambda$ and $\widehat{g}_t$ is fixed, we reduce the objective from maximizing TPR to minimizing $\lambda$. Consequently, the solution $\widehat{\lambda}_t$ from (Q1), together with $\widehat{g}_t$, satisfies the FPR constraint with high probability, i.e., $\mathrm{FPR}(\widehat{g}_t, \widehat{\lambda}_t) \leq \alpha$ (see Proposition 1 for more details).

At a high level, ASAT proceeds in two stages at each time step $t$. If the number of newly identified OOD samples $\Delta_t^{(ood)}$ exceeds the budget $\omega_{U_t}^{(ood)}$, we solve (P2) to update the scoring function $\widehat{g}_t$ and obtain a candidate threshold $\widehat{\lambda}'_t$; otherwise we retain $\widehat{g}_t = \widehat{g}_{t-1}$. In either case, the final threshold $\widehat{\lambda}_t$ is then computed via (Q1) to ensure FPR control. See Algorithm 1 and Appendix B for further details.

## 4 Theoretical Analysis

We provide a theoretical analysis of FPR control in stationary settings, where both $\mathcal{D}_0$ and $\mathcal{D}_1$ remain fixed over time (but unknown). Our goal is to show that ASAT maintains $\mathrm{FPR}(\widehat{g}_t, \widehat{\lambda}_t) \leq \alpha$ at all times. In our analysis, we use a *calibration-safe* variant of Algorithm 1, where FPR is estimated using *only samples collected after the currently active scoring function was trained*, removing the data dependence in calibration; we denote this $\widehat{\mathrm{FPR}}_t^{\mathrm{post}}$. Each newly trained scoring function is deployed only once (Q1) becomes feasible. See Algorithm 3 and Appendix C for full setup details and proofs.

**Proposition 1.** *Let $U_t$ be the total number of updates before time $t$, and let $\tau_t$ denote the training time of the active $\widehat{g}_t$. Let $N_t^{(o)} = \sum_{u=\tau_t+1}^{t} Z_u$, and $N_t^{(imp)}$ be the number of importance sampled OOD points in $(\tau_t, t]$. Define $\beta_t = N_t^{(imp)}/N_t^{(o)}$, $c_t = 1 + (1-p)\beta_t/p^2$, and $t_0 = \min\{u : c_u N_u^{(o)} \geq 173 \log(4/\delta)\}$. Let $\Lambda := \{\Lambda_{\min}, \Lambda_{\min} + \eta, \ldots, \Lambda_{\max}\}$, with discretization parameter $\eta > 0$. Then, w.p. at least $1 - \delta$, we have $\forall\, t \geq t_0$,*

*a)* $\displaystyle \sup_{\lambda \in \Lambda} \left| \widehat{\mathrm{FPR}}_t^{\mathrm{post}}(\widehat{g}_t, \lambda) - \mathrm{FPR}(\widehat{g}_t, \lambda) \right| \leq \psi_t(\delta)$, *where*

$$\psi_t(\delta) := \sqrt{\frac{3c_t}{N_t^{(o)}} \left[ 2 \log \log \left( \frac{3c_t N_t^{(o)}}{2} \right) + 2 \log \left( \frac{4U_t |\Lambda|}{\delta} \right) \right]}.$$

*b) Consequently, $\mathrm{FPR}(\widehat{g}_t, \widehat{\lambda}_t) \leq \alpha$.*

---

**Algorithm 1** ASAT (Adaptive Scoring Adaptive Threshold)

---

1: **procedure** ASAT($\alpha$, $p$, $\delta$, $\beta$, $\widehat{g}_0$, $\{\omega_i^{(ood)}\}_{i \geq 0}$, $X_0^{(id)}$)
2:   $\widehat{\lambda}_0 \leftarrow +\infty$, $X_0^{(ood)} \leftarrow \emptyset$, $U_0 \leftarrow 1$, $\Delta_0^{(ood)} \leftarrow 0$, upd $\leftarrow$ **false**
3:   **for** $t = 1, 2, \ldots$ **do**
4:     Observe $x_t$;  score $s_t \leftarrow \widehat{g}_{t-1}(x_t)$
5:     **if** $s_t \leq \widehat{\lambda}_{t-1}$ **then**                                                          $\triangleright$ $x_t$ is predicted as OOD
6:       $\ell_t \leftarrow 1$
7:     **else**                                                                                 $\triangleright$ $x_t$ is predicted as ID
8:       $\ell_t \sim \text{Bern}(p)$
9:     **if** $\ell_t = 1$ **then**
10:       $y_t \leftarrow \text{GetHumanLabel}(x_t)$                                              $\triangleright$ Obtain human label
11:       **if** $y_t = 0$ **then**                                                           $\triangleright$ $x_t$ is truly OOD
12:         $X_t^{(ood)} \leftarrow X_{t-1}^{(ood)} \cup \{x_t\}$;  $\Delta_t^{(ood)} \leftarrow \Delta_{t-1}^{(ood)} + 1$
13:       **else**
14:         $X_t^{(ood)} \leftarrow X_{t-1}^{(ood)}$;  $\Delta_t^{(ood)} \leftarrow \Delta_{t-1}^{(ood)}$
15:     **else**
16:       $X_t^{(ood)} \leftarrow X_{t-1}^{(ood)}$;  $\Delta_t^{(ood)} \leftarrow \Delta_{t-1}^{(ood)}$
17:
18:     $\triangleright$ Learn new scoring function
19:     **if** $\Delta_t^{(ood)} \geq \omega_{U_t}^{(ood)}$ **then**                        $\triangleright$ Enough new OOD samples are collected
20:       $\Delta_t^{(ood)} \leftarrow 0$;  upd $\leftarrow$ **false**
21:       $\widehat{g}_t, \widehat{\lambda}_t' \leftarrow \underset{g \in \mathcal{G}, \lambda \in \Lambda}{\arg\min} \left[ -\widehat{\text{TPR}}(g, \lambda) + \beta \widehat{\text{FPR}}_t(g, \lambda) \right]$      $\triangleright$ using $X_t^{(ood)}$ and $X_0^{(id)}$; see (P2)
22:       $\triangleright$ Estimate threshold
23:       $\widehat{\lambda}_t \leftarrow \underset{\lambda \in \Lambda}{\arg\min} \ \lambda$  s.t.  $\widehat{\text{FPR}}_t(\widehat{g}_t, \lambda) + \psi_t(\delta) \leq \alpha$      $\triangleright$ BinarySearch($X_t^{(ood)}$, $\widehat{g}_t$); see (Q1)
24:
25:       $\triangleright$ Model selection
26:       **if** $\widehat{\text{TPR}}(\widehat{g}_t, \widehat{\lambda}_t) - 2\zeta(\delta, |X_0^{(id)}|) > \widehat{\text{TPR}}(\widehat{g}_{t-1}, \widehat{\lambda}_{t-1})$ **then**      $\triangleright$ Selection criterion satisfied
27:         $U_t \leftarrow U_{t-1} + 1$;  upd $\leftarrow$ **true**                  $\triangleright$ Deploy new scoring function / threshold
28:       **else**
29:         $\widehat{g}_t \leftarrow \widehat{g}_{t-1}$;  $\widehat{\lambda}_t \leftarrow \widehat{\lambda}_{t-1}$;  $U_t \leftarrow U_{t-1}$      $\triangleright$ Retain old model / threshold
30:     **else**
31:       $U_t \leftarrow U_{t-1}$;  $\widehat{g}_t \leftarrow \widehat{g}_{t-1}$                              $\triangleright$ No model update
32:     **if not** upd **then**
33:       $\triangleright$ Estimate threshold
34:       $\widehat{\lambda}_t \leftarrow \underset{\lambda \in \Lambda}{\arg\min} \ \lambda$  s.t.  $\widehat{\text{FPR}}_t(\widehat{g}_t, \lambda) + \psi_t(\delta) \leq \alpha$      $\triangleright$ BinarySearch($X_t^{(ood)}$, $\widehat{g}_t$); see (Q1)
35:     **if** $\ell_t = 1$ **then**
36:       **Output** $y_t$                                                                $\triangleright$ True label via human feedback
37:     **else**
38:       **Output** $\widehat{y}_t := \mathbb{1}\{s_t > \widehat{\lambda}_{t-1}\}$                       $\triangleright$ Predicted label via one-sided threshold

---

**Interpretation.** Proposition 1 establishes the desired FPR control at all times in *calibration-safe* ASAT. It ensures that w.p. at least $1 - \delta$, a) the FPR estimate is within $\psi_t(\delta)$ of the true FPR for all $t$ and $\lambda$, and consequently b) FPR $\leq \alpha$ in ASAT. Importantly, this is a *safety* guarantee on FPR $\leq \alpha$. The TPR achievable depends on the separation between $\mathcal{D}_0$ and $\mathcal{D}_1$. Well-separated distributions yield TPR approaching 1. The more the overlap, the lesser the TPR at a given level of FPR. ASAT improves this separation in the scores over time by adapting scoring function using OOD feedback.

**Discussion.** Similar results were obtained for FSAT (which adapts thresholds for a *fixed* scoring function) using a similar UCB (Vishwakarma et al., 2024c). But, this approach does not extend directly to our setting, where ASAT also updates scoring functions and uses them for predictions and sample collections. Nevertheless, our design achieves comparable FPR control and higher TPR—since ASAT optimizes scoring functions to maximize TPR in (P2). However, these advantages come with a slight trade-off: $\psi_t(\delta)$ includes the additional $U_t$ term to ensure that it is also valid for time-varying $\widehat{g}_t$. In practice, we anticipate $U_t \ll N_t^{(o)}$, since updates occur only periodically. Thus, this extra dependence on $U_t$ results only in a minor increase in $\psi_t(\delta)$ from FSAT, which is offset by the improved TPR. Note that in the experiments, we use a heuristic variant $\psi_t^H(\delta)$ with tuned constants in place of the theoretical bound $\psi_t(\delta)$; see Section 5.1 and Appendix D.2 for details.

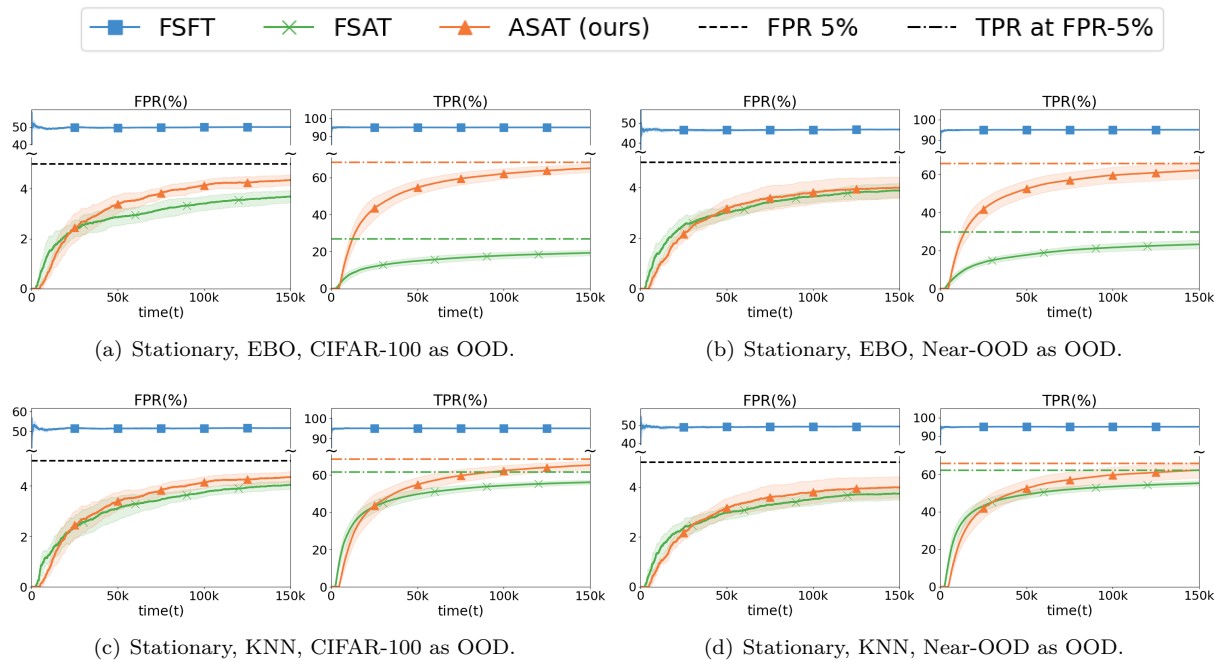

(a) Stationary, EBO, CIFAR-100 as OOD.

(b) Stationary, EBO, Near-OOD as OOD.

(c) Stationary, KNN, CIFAR-100 as OOD.

(d) Stationary, KNN, Near-OOD as OOD.

Figure 4: Results on the stationary setting for CIFAR-10 as ID, with each method repeated 5 times (mean and std shown). EBO and KNN are the initial scoring functions and are fixed for FSFT and FSAT. The dotted-dash lines represent the *expected* TPR at FPR-5% for EBO, KNN, and $\widehat{g}^*$ (matching colors).

We again emphasize that Proposition 1 applies to the calibration-safe variant described above, which differs from Algorithm 1 by restricting calibration to post-training samples (full details in Appendix C).

**Proof Sketch.** The proof of Proposition 1 entails challenges. First, the samples used to estimate the FPR in equation 2 are *dependent*. In particular, whether we receive the human label for $x_t$ depends on the previously learned scoring function $\widehat{g}_{t-1}$ and threshold $\widehat{\lambda}_{t-1}$, both of which are functions of previous data $\{x_1, ..., x_{t-1}\}$. This dependency prevents the direct application of the *i.i.d.* LIL bound (Howard and Ramdas, 2022). To overcome this, we leverage the martingale structure of our data and apply the martingale LIL bound (Khinchine, 1924; Balsubramani, 2015), yielding a time-uniform confidence sequence. Additionally, we ensure that these confidence intervals hold simultaneously for all $\lambda \in \Lambda$ and all deployed scoring functions by applying the union bound. See Appendix C for detailed proofs.

## 5 Experiments

We compare three methods: FSFT (fixed scoring functions and thresholds), FSAT (fixed scoring functions with adaptive thresholds), and our proposed ASAT (adaptive scoring functions and thresholds). We verify the following claims on OpenOOD benchmarks. For completeness, we also compare ASAT against a naive binary classifier in Appendix D.1.

**C1.** In stationary settings, ASAT (ours) $\succ$ FSAT $\succ$ FSFT, where $a \succ b$ means that method $a$ achieves better TPR than $b$ while maintaining FPR below $\alpha$, or that $b$ violates the FPR constraint.

**C2.** ASAT adapts to nonstationary OOD settings, updating scoring functions and thresholds to maximize TPR with *minimal* FPR violations.

### 5.1 Experiment Settings

**Datasets.** We use CIFAR-10 (Krizhevsky et al., 2009) as ID for the experiments. In stationary settings, we use CIFAR-100 and a mixture of CIFAR-100, Tiny-ImageNet (Deng et al., 2009), Places365 (Zhou et al., 2017) as OOD (referred to as *Near-OOD*). In nonstationary settings, we change the distribution at time

$t = 50\text{k}$ for all experiments. We use pairs of (1) CIFAR-100 and Places365 and (2) a mixture of MNIST (Deng, 2012), SVHN, and Texture (referred to as *Far-OOD*) and a mixture of CIFAR-100, Tiny-ImageNet, and Places365 (Near-OOD).

See Appendix D.7 and D.8 for additional experiments on different feature extractor (ViT-B/16 (Dosovitskiy et al., 2021)), ID datasets (CIFAR-100, ImageNet-1k (Deng et al., 2009)), OOD datasets, and scoring functions.

**Scoring Functions.** We use post-processing scoring functions based on a ResNet18 model (He et al., 2016), which is trained on CIFAR-10, from Open-OOD benchmarks: ODIN (Liang et al., 2018), Energy Score (EBO) (Liu et al., 2020), KNN (Sun et al., 2022), Mahalanobis Distances (MDS) (Lee et al., 2018a), and VIM (Wang et al., 2022). Both FSFT and FSAT used a fixed scoring function from the above benchmarks. For consistency, in ASAT, the same initial scoring function as FSFT is used until enough OOD samples are identified and the first update occurs.

**Evaluation FPR and TPR.** For evaluation, we count the actual number of false-positive and true-positive instances incurred by a method within a specific time frame. These *evaluation* FPR and TPR are defined as

$$
\begin{aligned}
\text{FPR}_t^{(eval)} &:= \frac{\sum_{u=t'}^{t} \mathbb{1}\{y_u = 0, \widehat{y}_u = 1\}}{\sum_{u=t'}^{t} \mathbb{1}\{y_u = 0\}}, \\
\text{TPR}_t^{(eval)} &:= \frac{\sum_{u=t'}^{t} \mathbb{1}\{y_u = 1, \widehat{y}_u = 1\}}{\sum_{u=t'}^{t} \mathbb{1}\{y_u = 1\}},
\end{aligned}
\tag{7}
$$

where $t' = t - N_w^{(eval)}$, and $N_w^{(eval)}$ denotes the window size of past instances used for evaluation.

**Choices of $\mathcal{G}$.** Our framework is flexible with the choice of $\mathcal{G}$. We use 2-layer ReLU neural networks that take a feature output from the penultimate layer of the Resnet18. To solve (P2), we use the Adam optimizer (Kingma and Ba, 2014). See Appendix D.3 for more details.

**Hyperparameters and Constants.** We perform a grid search to optimize our hyperparameters (See Appendix D.4). In particular, we set $\beta = 1.5$ and $\kappa = 50$. For the constants, we use $\alpha = 0.05$, $p = 0.2$, and $\gamma = 0.2$ throughout the main experiments. However, to further demonstrate the effectiveness of ASAT under different choices of the constants $\alpha$, $p$, and $\gamma$, we study the sensitivity of ASAT to $\alpha$, $p$, and $\gamma$ on synthetic Gaussian OOD and ID data in Appendix D.5.

**Empirical Convergence to Optimality.** We assess convergence of $(\widehat{g}_t, \widehat{\lambda}_t)$ by comparing $\text{TPR}_t^{(eval)}$ in equation 7 with the reference $\text{TPR}(\widehat{g}^\star, \lambda)$. Here, $(\widehat{g}^\star, \lambda)$ is near-optimal for (P1), where $\widehat{g}^\star$ is trained on *i.i.d.* OOD/ID data, and $\lambda$ is chosen to satisfy $\text{FPR}(\widehat{g}^\star, \lambda) = \alpha$. Intuitively, $\text{TPR}(\widehat{g}^\star, \lambda)$ represents the best achievable TPR under $\text{FPR} \leq \alpha$, and we expect $\text{TPR}_t^{(eval)}$ to converge toward it as $t$ increases in ASAT.

**Thresholds for Baselines.** While FSFT uses a fixed threshold at TPR-95%, both FSAT and ASAT update thresholds over time. For FSAT, we adopt the same heuristic UCB from Vishwakarma et al. (2024c). For ASAT we adapt the theoretical bound in equation 6 as

$$
\psi_t^H(\delta) := c_1 \sqrt{\frac{c_t}{N_t^{(o)}} \left[ \log\log\left( c_2 c_t N_t^{(o)} \right) + \log\left( \frac{c_3}{\delta} \right) \right]},
$$

with $c_1 = 0.65$, $c_2 = 0.75$, and $c_3 = 1.0$. Here, $c_2$ and $c_3$ are set as in Vishwakarma et al. (2024c), while $c_1$ is increased to account for the varying score functions (see Section 4). See Appendix D.2 for more details.

**Stationary Setting.** In this setting, $\mathcal{D}_0$ and $\mathcal{D}_1$ remain fixed. For evaluation, we set $N_w^{(eval)} = t$ (i.e., using all past data), since no abrupt changes in FPR/TPR are anticipated. For updating $\widehat{g}_t$, we set the optimization frequency $\omega_i^{(ood)}$ to 100 for the first 2k OOD points, 500 for the next 10k, and 1000 thereafter. This allows aggressive updates early on, with update frequency decreasing as $\widehat{g}_t$ becomes better trained. We analyze the effect of the update frequency $\omega^{(ood)}$ in Appendix D.6.

**Nonstationary Setting.** We manually change the OOD $\mathcal{D}_0$ at $t = 50k$, since variations in $\gamma$ or $\mathcal{D}_1$ do not impact the FPR. We set $N_w^{(eval)} = 50\text{k}$ for a more optimistic performance metric and use $\omega_i^{(ood)} = 100$ for the

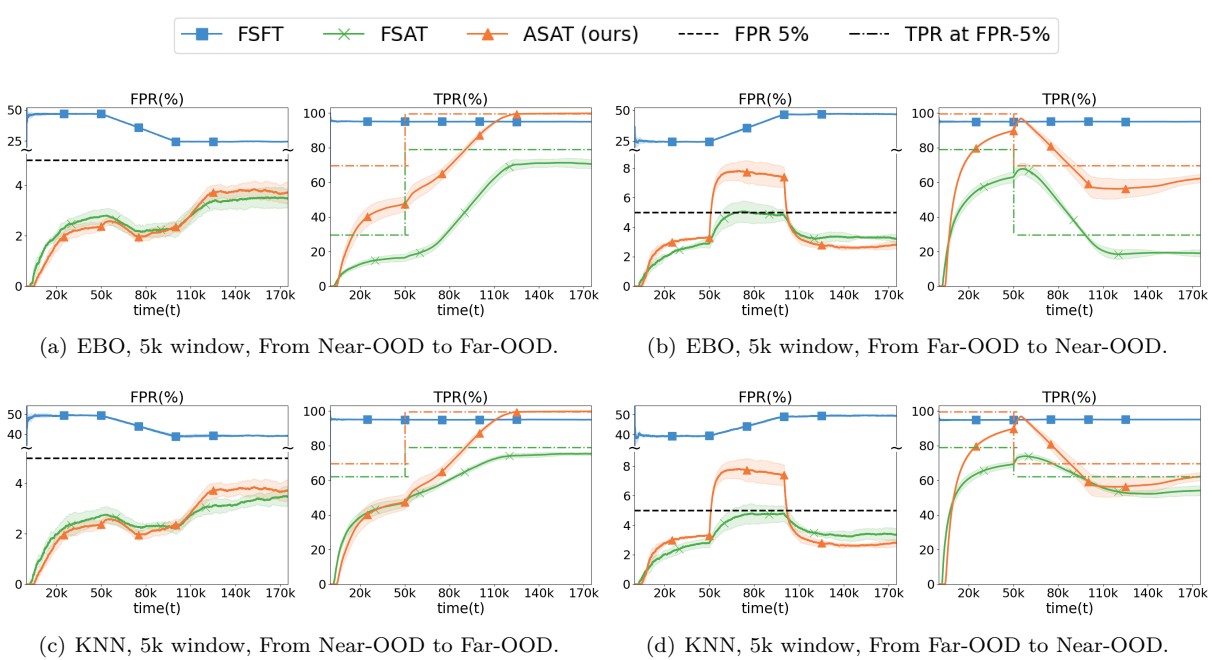

(a) EBO, 5k window, From Near-OOD to Far-OOD.  (b) EBO, 5k window, From Far-OOD to Near-OOD.

(c) KNN, 5k window, From Near-OOD to Far-OOD.  (d) KNN, 5k window, From Far-OOD to Near-OOD.

Figure 5: Results on the nonstationary setting, with each method repeated 5 times (mean and std shown). A shift happens between Near-OOD and Far-OOD mixtures at time $t = 50k$. EBO and KNN are the initial scoring functions and are fixed for FSFT and FSAT. The dotted-dash lines represent the TPR at FPR-5% for EBO, KNN, and $\widehat{g}^*$ (matching colors).

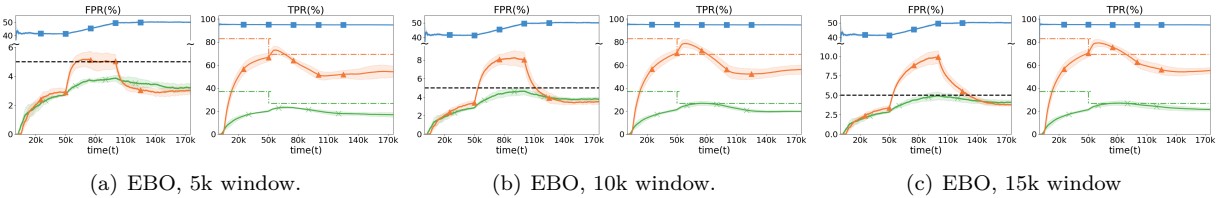

(a) EBO, 5k window.  (b) EBO, 10k window.  (c) EBO, 15k window

Figure 6: Results on the nonstationary setting with different window sizes 5k, 10k, and 15k, with each method repeated 5 times (mean and std shown). A shift happens from Places365 and CIFAR-100 at time $t = 50k$.

first 2k samples and 500 afterward to ensure $\widehat{g}_t$ adapts quickly to the new OOD. Since using all the past OOD samples (including *outdated ones*) becomes ineffective after a shift, we adopt a *windowed approach* that relies only on the most recent $N_w^{(est)}$ OOD samples for FPR estimation and updating $\widehat{g}_t$. This enables quicker adaptation to the new distribution. Note that the window size presents a trade-off: a smaller window encourages faster adaptation but results in a wider confidence interval, which in turn limits optimality.

*Remark.* All experiments use Algorithm 1 (all-history estimator Eq. 2, immediate deployment, heuristic UCB $\psi_t^H$), not the calibration-safe variant analyzed in Section 4. Results are thus empirical evidence for this practical variant, not a direct instantiation of Proposition 1.

## 5.2 Results and Discussions

**C1.** In stationary settings, ASAT aims to control FPR while maximizing TPR. In Figure 4, FSFT exhibits very high FPR at TPR-95%, whereas both FSAT and ASAT keep FPR consistently below 5% across all four settings. Initially, both FSAT and ASAT yield a TPR of 0% because insufficient OOD samples force $\widehat{\lambda}_t$ to be set to $\infty$ (as per (Q1)), until enough OOD samples are collected to satisfy $\psi_t(\delta) \leq \alpha$. Moreover, ASAT initially achieves a lower TPR than FSAT because it adapts both thresholds and scoring functions—resulting in a wider confidence bound $\psi_t(\delta)$ that requires more OOD samples—while FSAT updates only thresholds (see Section 4).

However, this early drawback is offset by its ability to maximize TPR later in deployment. ASAT quickly catches up with FSAT and eventually surpasses the best achievable TPR by FSAT at FPR-5% in all settings. In particular, when both methods start with EBO, ASAT shows over a 40% improvement in TPR towards the end (see Figures 4(a) and 4(b)). As expected, $\text{TPR}(\widehat{g}_t, \widehat{\lambda}_t)$ converges to $\text{TPR}(\widehat{g}^\star, \lambda)$, confirming optimality. These verify ASAT effectively updates both scoring functions and thresholds to maximize TPR while ensuring FPR control, outperforming the baselines.

**C2.** In nonstationary settings, ASAT aims to maximize TPR while minimizing FPR violation if it occurs. As shown in Figure 5, FPR violations occur when the new OOD is less distinguishable from the ID than the old OOD (e.g., Far-OOD to Near-OOD in Figures 5(b) and 5(d)) whereas

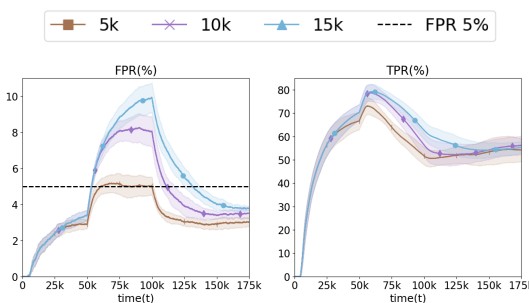

Figure 7: Results on nonstationary OOD from Places365 to CIFAR-100 with different window $N_w^{(est)}$; only ASAT results are shown.

no FPR violations are observed in the reverse shift (Figures 5(a) and 5(c)). In Figure 6, ASAT initially shows larger FPR violations than FSAT, which is expected since it adapts (or *specializes*) the scoring functions based on previous OOD data. However, ASAT quickly reduces FPR below $\alpha$ and effectively adapts to the new OOD, consistently achieving higher TPR than FSAT both before and after the shift. Figure 7 illustrates smaller windows enable faster adaptation and reduce FPR violations, though they limit the TPR convergence.

## 6 Related Works

**OOD Detection.** Ensuring robustness to out-of-distribution (OOD) inputs has been an important topic in the machine learning community, especially when deploying systems in safety-critical domains (Salehi et al., 2022; Yang et al., 2024; Bendale and Boult, 2015). A wide range of post-hoc and training-time methods have been proposed for OOD detection, typically by designing scoring functions that assign confidence values to quantify the uncertainty of whether inputs are OOD. The maximum softmax probability was shown to provide a simple but effective baseline (Hendrycks and Gimpel, 2017). Building on this idea, different methods have been proposed to improve scoring. Energy-based approaches exploit the connection between energy and likelihood (Liu et al., 2020), while Mahalanobis distance–based methods assume Gaussian class-conditional distributions and measure distances from ID clusters (Lee et al., 2018b). Self-supervised approaches (Sehwag et al., 2021) learn effective representations without requiring additional labels. Another direction uses nonparametric similarity in the feature space, for example by applying deep nearest neighbors (Sun et al., 2022). Hybrid approaches combine different sources of information, such as virtual logit matching (Wang et al., 2022). A rich line of work also focuses on calibration, since overconfident neural networks can undermine the reliability of thresholding. Confidence learning explicitly trains models to output calibrated scores (DeVries and Taylor, 2018), while logit normalization reduces overconfidence by rescaling logits (Wei et al., 2022). More recent work has proposed adversarial reciprocal points (Chen et al., 2021), hyperspherical embeddings (Ming et al., 2023), and large semantic space detectors (Huang and Li, 2021). However, these methods often rely only on in-distribution (ID) data to design scoring functions and set static score thresholds, focusing on maximizing TPR. This often leads to high FPRs, since they are not explicitly controlled. Furthermore, having fixed scoring functions and thresholds may limit a system's ability to adapt to new OOD inputs under distribution shifts.

Recent works have explored offline strategies that incorporate OOD data during training. Katz-Samuels et al. (2022) train classifiers and OOD detectors jointly on unlabeled mixtures of ID and OOD data, while Bai et al. (2024) use active learning to obtain a small number of labeled OOD samples. A complementary approach (Magesh et al., 2023) develops a statistically rigorous framework based on multiple hypothesis testing and conformal $p$-values, which provides conditional false alarm guarantees. However, these methods are designed for offline use and do not adapt once deployment begins. In contrast, our goal is to develop an online framework that can adjust to new OOD variations on the fly while maintaining strict control of FPRs.

**FPR Control for OOD Detection.** Explicit control of the FPR in OOD detection has recently begun to receive systematic attention. Vishwakarma et al. (2024c) introduced a framework (referred as FSAT

here) that adapts thresholds over time while holding the scoring function fixed, thereby guaranteeing that FPR remains below a user-specified level throughout deployment. This method is an important step toward making OOD detection more reliable in practice, as it directly addresses the lack of FPR guarantees in many earlier approaches. However, FSAT does not update the scoring function itself. As a result, the achievable TPR is inherently capped, since a fixed score may perform well on some OOD inputs but fail on others. Our work extends this line of research by jointly updating both the scoring function and the threshold on the fly, enabling the system to explicitly maximize TPR subject to an FPR constraint. Related ideas have also been studied in model-assisted data labeling (Vishwakarma et al., 2023; 2024b;a; 2025), where thresholds and model's confidence scores are adapted to improve the quality of auto-labeling, and in conformal prediction (Stutz et al., 2022), where thresholds are optimized to balance coverage and accuracy.

**Time-uniform Confidence Sequences.** Our theoretical framework builds on the rich literature of time-uniform confidence sequences, which provide concentration bounds that remain valid at all times. Classical results, such as the law of the iterated logarithm (Darling and Robbins, 1967; Lai, 1976), established some of the earliest tools for sequential inference. More recent works have extended these ideas to modern problems in bandits and active learning, including algorithms for exploration and sequential quantile estimation (Jamieson et al., 2014; Howard and Ramdas, 2022), and these techniques have also found applications in ranking and clustering tasks (Heckel et al., 2019; Vinayak, 2018; Chen et al., 2023). However, these results rely on i.i.d. assumptions of data, which do not hold in our setting. In our theoretical analysis, we use importance sampling, combined with time-varying scoring functions and thresholds, which introduces data dependencies that invalidate standard concentration bounds. To overcome this challenge, we draw on martingale-based extensions of the law of the iterated logarithm (Balsubramani, 2015), which provide finite-time guarantees under such dependence. By constructing suitable martingale sequences, we are able to derive confidence sequences that remain valid even when the scoring function evolves over time. This adaptation is essential for providing rigorous FPR guarantees in online OOD detection systems that continuously learn from new data.

## 7 Conclusion

We introduced a human-in-the-loop OOD detection framework that updates both scoring functions and thresholds to maximize TPR while strictly controlling FPRs. Our approach formulated this as a constrained joint optimization problem, with a relaxed version developed for practical learning. To guarantee FPR control throughout deployment, we designed novel time-uniform confidence sequences that account for dependencies from importance sampling and time-varying scores and thresholds. Empirical evaluations show that our approach outperforms baselines by maintaining FPR control and improving TPR.

## 8 Limitations and Future Work

While ASAT provides a new adaptive OOD detection framework with formal FPR control guarantees, several limitations remain and motivate future work.

**Noisy Human Feedback.** In this work, we assume *expert human labels*, as would be expected from domain specialists. While this is reasonable in the safety-critical settings motivating our work, there are settings where these inputs could be noisy. The method can be extended to handle noisy feedback using techniques from the label noise literature (Natarajan et al., 2013; Chen et al., 2023; Ibrahim et al., 2025). Incorporating such corrections into FPR estimation and extending FPR guarantees under noisy labels is a promising direction for future work.

**Shift Detection for Nonstationary Settings.** Our guarantees assume stationary OOD distributions. A possible extension is to use change-point detection (e.g., by monitoring FPR) to detect shifts and re-initialize ASAT, which allows the stationary guarantees to apply piecewise across each stationary interval. A full theoretical analysis under nonstationary settings is left for future work.

**Different Types of Distribution Shifts.** ASAT focuses on semantic OOD shifts, but covariate shifts (e.g., noise, resolution changes) are also important. Evaluating and extending ASAT under such shifts is an open area for future research.

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

## Appendix

The appendix is organized as follows. We provide a summary of the variables and constants in Table 2. Then, we give an algorithm for the entire ASAT workflow in Section B and detailed proofs for the FPR control in Section C. In Section D, we provide additional experimental results and details of the experiment protocol and hyperparameters used for the experiments.

## A  Glossary

We provide the following list of variables and constants used in our framework (Table 2).

## B  Additional Details for Methodology

We provide a detailed description of the ASAT workflow in Algorithm 1. We also provide the algorithm for binary search procedures used to solve (Q1) in Algorithm 2.

---

**Algorithm 2** Binary Search to Solve (Q1)

---

 1: **procedure** BINARYSEARCH($X_t^{(ood)}, \widehat{g}_t$)
 2:     $\lambda_l = \Lambda_{min}, \lambda_h = \Lambda_{max}, feasible = \texttt{False}, \epsilon \approx 0$
 3:     **while** $\lambda_h - \lambda_l > \epsilon$ **do**
 4:         $\lambda_m = (\lambda_l + \lambda_h)/2$
 5:         **if** $\widehat{\mathrm{FPR}}_t(\widehat{g}_t, \lambda_m) + \psi_t(\delta) \leq \alpha$ **then**
 6:             $feasible = \texttt{True}$
 7:             $\lambda_h = \lambda_m$
 8:         **else**
 9:             $\lambda_l = \lambda_m$
10:     **return** $\lambda_m, feasible$

---

| Symbol | Definition/Description |
|---|---|
| $\mathcal{X}$ | feature space. |
| $\mathcal{Y}$ | label space $\{0, 1\}$, 0 for OOD and 1 for ID. |
| $\mathcal{D}_0, \mathcal{D}_1$ | the underlying distributions of OOD and ID points |
| $\gamma$ | mixture ratio of OOD and ID distributions. |
| $x_t, s_t, y_t, \widehat{y_t}$ | sample, score, true label, and predicted label at time $t$. |
| $X_t^{(ood)}$. | *dependent* OOD samples collected up to time $t$. |
| $X_0^{(id)}$ | i.i.d. ID samples available before deployment. |
| $\mathrm{FPR}(g, \lambda)$ | population level false positive rate defined by scoring function $g$ and threshold $\lambda$. |
| $\mathrm{TPR}(g, \lambda)$ | population level true positive rate defined by scoring function $g$ and threshold $\lambda$. |
| $\widehat{\mathrm{FPR}}_t(g, \lambda)$ | estimated FPR at time $t$, adjusted to account for importance sampling (see equation 2). |
| $\widehat{\mathrm{TPR}}(g, \lambda)$ | estimated TPR fixed for all $t$, based on i.i.d. ID points (see equation 3). |
| $\widetilde{\mathrm{FPR}}_t(g, \lambda)$ | smoothed $\widehat{\mathrm{FPR}}_t(g, \lambda)$ (see equation 4) used for optimization in (P2) |
| $\widetilde{\mathrm{TPR}}(g, \lambda)$ | smoothed $\widehat{\mathrm{TPR}}(g, \lambda)$ (see equation 5) used for optimization in (P2). |
| $\mathrm{FPR}_t^{(eval)}$ | evaluation FPR for experiments in equation 7. |
| $\mathrm{TPR}_t^{(eval)}$ | evaluation TPR for experiments in equation 7. |
| $\widehat{\mathrm{FPR}}_t^{\mathrm{post}}(g, \lambda)$ | post-training FPR estimate for the calibration-safe variant (see equation 10). |
| $\mathcal{G}, \Lambda$ | hypothesis class of scoring functions over $\mathcal{X}$ and space of thresholds. |
| $(g^*, \lambda^*)$ | the optimal solution (non-unique) to the joint optimization problem in (P1). |
| $(\widehat{g}_t, \widehat{\lambda}_t)$ | scoring function and threshold learned at time $t$. |
| $\widehat{\lambda}'_t$ | *unsafe* threshold obtained from (P1) together with $\widehat{g}_t$. |
| $i_u$ | indicator of whether $x_t$ was importance sampled. |
| $\sigma(\kappa, z)$ | sigmoid function for input $z$. |
| $\omega_i^{(ood)}$ | optimization frequency, indicating the number of new OOD points required for $i$th updating. |
| $\beta_t, c_t$ | time-dependent variables used for LIL-based confidence bounds (see Proposition 1). |
| $\Delta_t^{(ood)}$ | number of *new* OOD points identified via human feedback up to time $t$ since *last update*. |
| $N_t^{(imp)}$ | number of OOD points identified via importance sampling up to time $t$. |
| $U_t$ | number of total deployments of new scoring functions by time $t$. |
| $N_w^{(eval)}$ | window size that the system uses for computing evaluation FPR and TPR. |
| $N_w^{(est)}$ | window size for estimation and optimization for nonstationary OOD settings. |
| $p$ | importance sampling probability. |
| $\delta$ | failure probability for confidence intervals. |
| $\alpha$ | the threshold for FPR for all time points. |
| $\eta$ | discretization parameter for $\Lambda$. |
| $\beta$ | optimization parameter that controls a trade-off between TPR and FPR. |
| $\kappa$ | the smoothness parameter of sigmoid functions |
| $(c_1, c_2, c_3)$ | constants use for heuristic LIL bounds. |
| $\psi_t(\delta)$ | LIL-based confidence interval at time $t$. |
| $\psi_t^H(\delta)$ | LIL-based heuristic interval at time $t$. |
| $\zeta(\delta, n)$ | DKW confidence interval for TPR. |

Table 2: Glossary of variables and symbols used in the framework.

**DKW Criterion for Model Selection.** Suppose an update occurs at time $t$ (i.e., $\Delta_t^{ood} \geq \omega_{U_t}^{ood}$) by solving (P2). We ensure that the updated scoring function $\widehat{g}_t$ outperforms $\widehat{g}_{t-1}$, by applying following TPR criterion:

$$\widehat{\mathrm{TPR}}(\widehat{g}_t, \widehat{\lambda}_t) - 2\zeta(\delta, |X_0^{(id)}|) > \widehat{\mathrm{TPR}}(\widehat{g}_{t-1}, \widehat{\lambda}_{t-1}),$$

where $\zeta(\delta, |X_0^{(id)}|)$ is the Dvoretzky–Kiefer–Wolfowitz (DKW) confidence interval (Dvoretzky et al., 1956) given by

$$\zeta(\delta, |X_0^{(id)}|) = \sqrt{\frac{1}{|X_0^{(id)}|} \log\left(\frac{2}{\delta}\right)},$$

which is valid w.p. at least $1 - \delta$. This confidence bound relies on the fact that $X_0^{(id)}$ is i.i.d. and the TPR can be expressed as $\mathrm{TPR}(g, \lambda) = 1 - F_1(\lambda; g)$, where $F_1$ is the CDF of the ID score distribution mapped by $g$.

## C  Proofs

**Summary of ASAT Settings.**   At each time $t$, the ASAT system processes an input $x_t$ using scoring function $\widehat{g}_{t-1}$ and threshold $\widehat{\lambda}_{t-1}$, both learned at the previous time $t-1$. It first computes score $s_t = \widehat{g}_{t-1}(x_t)$ for $x_t$. If $s_t \leq \widehat{\lambda}_{t-1}$, then $x_t$ is considered OOD and receives a human label $y_t \in \{0, 1\}$. If $s_t > \widehat{\lambda}_{t-1}$, $x_t$ is considered ID, and in this case, we obtain the human label $y_t$ only with probability $p \in (0, 1)$ (i.e., importance sampling). Now, if $x_t$ is identified as OOD and the system has collected enough new OOD points required for the next update (i.e., $\Delta_{U_t}^{(ood)}$), we estimate the FPR and TPR in equation 1 using the collected finite samples to solve the optimization in (P2). The FPR and TPR estimates are defined as

$$\widehat{\text{FPR}}_t(g, \lambda) := \frac{\sum_{u=1}^{t} Z_u \cdot \mathbb{1}\{g(x_u) > \lambda\}}{\sum_{u=1}^{t} Z_u}, \tag{8}$$

$$Z_u := \begin{cases} 1 & \text{if } s_u \leq \widehat{\lambda}_{u-1}, i_u = 0, y_u = 0, \\ \frac{1}{p} & \text{if } s_u > \widehat{\lambda}_{u-1}, i_u = 1, y_u = 0, \\ 0 & \text{otherwise}, \end{cases}$$

and

$$\widehat{\text{TPR}}(g, \lambda) := \frac{1}{|X_0^{(id)}|} \sum_{x \in X_0^{(id)}} \mathbb{1}\{g(x) > \lambda\}. \tag{9}$$

In (P2), the differentiable surrogates of the FPR and TPR estimates from equation 4 and equation 5 are used for efficient optimization. If an update occurs, the system obtains a new scoring function $\widehat{g}_t$ and threshold $\widehat{\lambda}_t'$. Otherwise, the previous scoring function is retained, i.e., $\widehat{g}_t = \widehat{g}_{t-1}$. Since the optimization problem in (P2) is unconstrained, there's no guarantee that the original FPR constraint will be satisfied by $\widehat{g}_t$ and $\widehat{\lambda}_t'$. Therefore, the system solves a different optimization in (Q1) to find $\widehat{\lambda}_t$, replacing $\widehat{\lambda}_t'$, based on the current scoring function $\widehat{g}_t$. In this optimization, we use the FPR estimate in equation 2 and LIL-based upper confidence bounds (UCB) to ensure that the true FPR constraint is satisfied with high probability. Additional details on the workflow are provided in B. This is summarized precisely in Algorithm 1.

**Calibration-Safe Variant of Algorithm 1.**  For theoretical analysis, we consider a modification to Algorithm 1 in how the threshold is calibrated for each new scoring function. The formal FPR guarantee in Proposition 3 holds rigorously under adaptive scoring for this calibration-safe variant. We denote the $k$-th scoring function as $\widehat{g}^{(k)}$, trained at time $\tau_k$ using OOD data collected up to $\tau_k$. Observe when $\widehat{g}^{(k)}$ is trained at $\tau_k$, it may depend on samples $x_u$ for $u \leq \tau_k$ through training in (P2).

To avoid this explicit dependency of the estimate of FPR for $\widehat{g}^{(k)}$ on the samples it has been trained on, we slightly modify Algorithm 1 in how thresholds are calibrated: for each $\widehat{g}^{(k)}$ trained at time $\tau_k$, the threshold is calibrated using only post-training samples ($u > \tau_k$). To avoid a cold-start ($\lambda = +\infty$) after each scoring function update, deployment of $\widehat{g}^{(k)}$ is delayed: the previous $\widehat{g}^{(k-1)}$ remains active with its already-calibrated threshold, while $\widehat{g}^{(k)}$ is calibrated in the background. Once (Q1) becomes feasible for $\widehat{g}^{(k)}$, it is deployed with its calibrated threshold. FPR control is maintained throughout: by $\widehat{g}^{(k-1)}$ during the waiting period, and by $\widehat{g}^{(k)}$ afterward. Formally, for any scoring function $\widehat{g}^{(k)}$ trained at time $\tau_k$ and any $t > \tau_k$, we define the post-training FPR estimator as:

$$\widehat{\text{FPR}}_t^{\text{post}}(\widehat{g}^{(k)}, \lambda) := \frac{\sum_{u=\tau_k+1}^{t} Z_u \mathbf{1}\{\widehat{g}^{(k)}(x_u) > \lambda\}}{\sum_{u=\tau_k+1}^{t} Z_u}. \tag{10}$$

Intuitively, this estimator excludes all samples that existed before training to eliminate the dependence. The theoretical result below assumes this workflow and FPR estimator.

**Remark 1.** *In all experiments we use Algorithm 1 as presented, with the all-history FPR estimator from Eq. 8 and the heuristic UCB constants from Appendix D.2. That is, in practice, the threshold calibration is done without sample splitting. Understanding why double dipping into the samples used to train the new scoring function for calibration works well in practice is an interesting future research direction.*

---

**Algorithm 3** Calibration-Safe Variant for Theoretical Analysis

---

1: Data collection, importance sampling, and scoring-function training via (P2) proceed as in Algorithm 1.
2: When a new scoring function $\widehat{g}^{(k)}$ is trained at time $\tau_k$, keep the previous calibrated scoring function active.
3: Calibrate $\widehat{g}^{(k)}$ in the background using only post-training samples $u > \tau_k$:

$$\widehat{\mathrm{FPR}}_t^{\mathrm{post}}(\widehat{g}^{(k)}, \lambda) = \frac{\sum_{u=\tau_k+1}^t Z_u \mathbf{1}\{\widehat{g}^{(k)}(x_u) > \lambda\}}{\sum_{u=\tau_k+1}^t Z_u}.$$

4: At each time $t > \tau_k$, solve (Q1) for $\widehat{g}^{(k)}$ using $\widehat{\mathrm{FPR}}_t^{\mathrm{post}}$ and the corresponding post-training confidence bound.
5: Deploy $\widehat{g}^{(k)}$ only once (Q1) is feasible; otherwise keep the previous calibrated scoring function active.

---

**Proof Outline.** Our goal in ASAT is to ensure that $\mathrm{FPR}(\widehat{g}_t, \widehat{\lambda}_t)$ remains below $\alpha$ at all times. Similar results were achieved for FSAT, where thresholds were adapted for a *fixed* scoring function (Vishwakarma et al., 2024c). Their results relied on constructing time-uniform FPR estimation bounds for all times $t$ and thresholds $\lambda \in \Lambda$. The OOD samples used to estimate the FPR are non-i.i.d., as the decision to receive a human label for $x_t$ depends on the previously learned $\widehat{g}_{t-1}$ and $\widehat{\lambda}_{t-1}$, which themselves are functions of past data $\{x_1, \ldots, x_{t-1}\}$. In FSAT, this dependency is handled by applying the martingale version of the LIL bound (Khinchine, 1924; Balsubramani, 2015), which leads to a time-uniform confidence sequence. However, in ASAT, the same LIL-based UCB does not directly apply, as the system periodically adapts new, *time-varying*, scoring functions and uses them for predictions and sample collection. Despite this, we claim that our design still ensures comparable FPR control under the above modifications. We first show that each $x_t$ identified as OOD is an unbiased estimator for the FPR.

**Lemma 1.** *Let $Z_u$ be defined as in equation 2, and let $i_u$ be the indicator for whether $x_u$ is importance sampled. Given $g \in \mathcal{G}$ and $\lambda \in \Lambda$, we have,*

*a)* $\mathbb{E}_{x_u, i_u}[Z_u \mathbf{1}\{g(x_u) > \lambda\}] = \gamma \cdot \mathrm{FPR}(g, \lambda),$
*b)* $\mathbb{E}_{x_u, i_u}[Z_u] = \gamma.$

*Consequently, we have*

$$\mathbb{E}_{x_u, i_u}[Z_u(\mathbf{1}\{g(x_u) > \lambda\} - \mathrm{FPR}(g, \lambda))] = 0.$$

*Proof.* Since $x_u \sim \gamma \mathcal{D}_0 + (1-\gamma)\mathcal{D}_1$ and $Z_u = 0$ whenever $y_u = 1$, we condition on $y_u$:

$$\mathbb{E}_{x_u, i_u}[Z_u \mathbf{1}\{g(x_u) > \lambda\}] = \gamma \cdot \mathbb{E}_{x_u, i_u}[Z_u \mathbf{1}\{g(x_u) > \lambda\} \mid y_u = 0],$$

since $Z_u = 0$ when $y_u = 1$. It remains to evaluate the conditional expectation. Conditioned on $y_u = 0$ we have $x_u \sim \mathcal{D}_0$. Now, set $m_u := \mathbb{P}(\widehat{g}_{u-1}(x_u) \leq \widehat{\lambda}_{u-1} \mid x_u, y_u = 0)$. Then, we have

$$\mathbb{E}_{x_u, i_u}[Z_u \mathbf{1}\{g(x_u) > \lambda\} \mid y_u = 0] = \mathbb{E}_{x_u \sim \mathcal{D}_0}\left[\mathbf{1}\{g(x_u) > \lambda\} \mathbb{E}_{i_u \mid x_u, y_u = 0}[Z_u]\right].$$

Conditioned on $x_u$ additionally, we have

$$\mathbb{E}_{i_u \mid x_u, y_u = 0}[Z_u] = m_u \cdot \mathbb{P}(i_u = 0 \mid \widehat{g}_{u-1}(x_u) \leq \widehat{\lambda}_{u-1}) + \frac{1}{p}(1 - m_u) \cdot \mathbb{P}(i_u = 1 \mid \widehat{g}_{u-1}(x_u) > \widehat{\lambda}_{u-1})$$

$$= m_u + \frac{1}{p}(1 - m_u)p = 1.$$

Hence, we have

$$\mathbb{E}_{x_u, i_u}[Z_u \mathbf{1}\{g(x_u) > \lambda\} \mid y_u = 0] = \mathbb{E}_{x_u \sim \mathcal{D}_0}[\mathbf{1}\{g(x_u) > \lambda\}] = \mathrm{FPR}(g, \lambda).$$

Combining gives (a). Part (b) follows identically with $\mathbf{1}\{g(x_u) > \lambda\}$ replaced by 1. The consequence then follows directly from (a) and (b) by linearity. $\square$

Next, we identify a martingale structure within the OOD samples. This step is crucial, as it will allow us to apply the martingale-LIL, yielding a time-uniform confidence sequence.

**Lemma 2.** *Fix any $g \in \mathcal{G}$, $\lambda \in \Lambda$, and starting index $s \geq 1$. For any time step $t \geq s$, define*

$$M_t^{(s)}(g, \lambda) := \sum_{u=s}^{t} Z_u \Big( \mathbb{1}\{g(x_u) > \lambda\} - \mathrm{FPR}(g, \lambda) \Big).$$

*Then, the stochastic process $\{M_t^{(s)}(g, \lambda)\}_{t \geq s}$ is a martingale with respect to the filtration $\{\mathcal{F}_t\}_{t \geq s}$, where $\mathcal{F}_t$ is the $\sigma$-field generated by the events until time $t$.*

*Proof.* We observe that $\mathbb{E} M_t^{(s)} < \infty$ and $M_t^{(s)}$ is $\mathcal{F}_t$-measurable. Now, we have for any $t > s$,

$$\mathbb{E}[M_t^{(s)}(g, \lambda) \mid \mathcal{F}_{t-1}] = \mathbb{E}\Big[ Z_t(\mathbb{1}\{g(x_t) > \lambda\} - \mathrm{FPR}(g, \lambda)) + M_{t-1}^{(s)}(g, \lambda) \mid \mathcal{F}_{t-1} \Big]$$
$$= \mathbb{E}\Big[ Z_t(\mathbb{1}\{g(x_t) > \lambda\} - \mathrm{FPR}(g, \lambda)) \Big] + M_{t-1}^{(s)}(g, \lambda)$$
$$= M_{t-1}^{(s)}(g, \lambda),$$

where the third equality follows from Lemma 1 applied conditionally on $\mathcal{F}_{t-1}$. □

In the calibration-safe variant, we apply Lemma 2 with $g = \widehat{g}^{(k)}$ and $s = \tau_k + 1$, so that $\widehat{g}^{(k)}$ is fixed relative to all samples in the sum (see Proposition 3). Lemma 2 leads to the following proposition on the LIL-based time-uniform confidence intervals for $\{M_t(g, \lambda)\}$.

**Proposition 2.** *Fix $g \in \mathcal{G}$ and a starting index $s \geq 1$. For any $t \geq s$, let $N_t^{(o)} = \sum_{u=s}^{t} Z_u$ and $N_t^{(imp)}$ be the number of importance sampled OOD points in $[s, t]$ and $\beta_t = \frac{N_t^{imp}}{N_t^{(o)}}$. Let $c_t := 1 + \frac{1-p}{p^2}\beta_t$, and $t_0 = \min\{u > s : c_u N_u^{(o)} \geq 173 \log(4/\delta)\}$. Let $M_t^{(s)}$ be defined as in Lemma 2. Let $\Lambda := \{\Lambda_{\min}, \Lambda_{\min} + \eta, ...., \Lambda_{\max}\}$, with discretization parameter $\eta > 0$. Then, we have*

$$\mathbb{P}\left( \forall\, t \geq t_0 : \sup_{\lambda \in \Lambda} |M_t^{(s)}(g, \lambda)| < \psi'(t, \delta) \right) \geq 1 - \delta,$$

*where*

$$\psi_t'(\delta) := \sqrt{3 c_t N_t^{(o)} \left[ 2 \log \log \left( \frac{3 c_t N_t^{(o)}}{2} \right) + \log \left( \frac{2|\Lambda|}{\delta} \right) \right]}.$$

*Proof.* For the martingale sequence $\{M_t^{(s)}(g, \lambda)\}_{t > s}$ defined in Lemma 2, the following is true for all $t > s$,

$$|M_t^{(s)} - M_{t-1}^{(s)}| \leq \begin{cases} 1 & : s_u \leq \widehat{\lambda}_{u-1}, i_u = 0, y_u = 0 \\ 1/p & : s_u > \widehat{\lambda}_{u-1}, i_u = 1, y_u = 0 \\ 0 & : o.w. \end{cases}$$

Hence, we know that $\{M_t^{(s)}\}$ has a bounded increment for all $t$. Define now $N_t^{(o)} = \sum_{u=s}^{t} Z_u$ and $N_t^{(imp)}$ be the number of importance sampled OOD points in $[s, t]$. Set $\beta_t = \frac{N_t^{imp}}{N_t^{(o)}}$, $c_t := 1 + \frac{1-p}{p^2}\beta_t$, and $t_0 = \min\{u > s : c_u N_u^{(o)} \geq 173 \log(4/\delta)\}$. Then, applying the Martingale LIL result from Balsubramani (2015), we obtained the following uniform concentration for a given $g \in \mathcal{G}$ and $\lambda \in \Lambda$:

$$\mathbb{P}\left( \forall\, t \geq t_0 : |M_t^{(s)}(g, \lambda)| \leq \sqrt{3 c_t N_t^{(o)} \left[ 2 \log \log \left( \frac{3 c_t N_t^{(o)}}{2} \right) + \log \left( \frac{2|\Lambda|}{\delta} \right) \right]} \right) \geq 1 - \delta.$$

The result follows by applying the union bound over the discretized class $\Lambda$ of $\lambda$. □

Proposition 2 provides a time-uniform confidence sequence valid for all $t \geq t_0$ and $\lambda \in \Lambda$, but it holds for a fixed $g \in \mathcal{G}$ and must be adapted to account for *time-varying* scoring functions $g_t$ for all $t \geq t_0$. Under the calibration-safe variant, the following proposition gives the desired result.

**Proposition 3** (Restatement of Proposition 1). *Let $U_t$ be the number of scoring function updates before time $t$, and let $\tau_t$ denote the training time of the active scoring function $\widehat{g}_t$. Applying Proposition 2 with $s = \tau_t + 1$, define $N_t^{(o)} = \sum_{u=\tau_t+1}^{t} Z_u$, $N_t^{(imp)}$ as the number of importance-sampled OOD points in $(\tau_t, t]$, $\beta_t = N_t^{(imp)}/N_t^{(o)}$, $c_t = 1 + (1-p)\beta_t/p^2$, and $t_0 = \min\{u : c_u N_u^{(o)} \geq 173\log(4/\delta)\}$. Let $\Lambda := \{\Lambda_{\min}, \Lambda_{\min} + \eta, \ldots, \Lambda_{\max}\}$, with discretization parameter $\eta > 0$. Then,*

$$\mathbb{P}\left(\forall\, t \geq t_0 : \sup_{\lambda \in \Lambda} \left|\widehat{\mathrm{FPR}}_t^{\,\mathrm{post}}(\widehat{g}_t, \lambda) - \mathrm{FPR}(\widehat{g}_t, \lambda)\right| < \psi_t(\delta)\right) \geq 1 - \delta,$$

*where*

$$\psi_t(\delta) := \sqrt{\frac{3c_t}{N_t^{(o)}}\left[2\log\log\left(\frac{3c_t N_t^{(o)}}{2}\right) + 2\log\left(\frac{4U_t|\Lambda|}{\delta}\right)\right]}.$$

*Consequently, under the calibration-safe variant of ASAT, w.p. at least $1 - \delta$, we have, for all $t \geq t_0$,*

$$\mathrm{FPR}(\widehat{g}_t, \widehat{\lambda}_t) \leq \alpha.$$

*Proof.* We wish the bound from Proposition 2 to hold simultaneously for every scoring function that ASAT deploys under the calibration-safe variant. For the $k$-th deployed scoring function $\widehat{g}^{(k)}$, trained at time $\tau_k$: conditional on $\mathcal{F}_{\tau_k}$, $\widehat{g}^{(k)}$ is a fixed element of $\mathcal{G}$, and for $u > \tau_k$, $x_u$ is independent of $\widehat{g}^{(k)}$ given $\mathcal{F}_{u-1}$ (since $\mathcal{F}_{\tau_k} \subseteq \mathcal{F}_{u-1}$). Setting $s = \tau_k + 1$, the conditions of Lemma 1, Lemma 2, and Proposition 2 are satisfied with $g = \widehat{g}^{(k)}$ and the newly defined post-training quantities $N_t^{(o)}$, $c_t$ over $(\tau_k, t]$. The $k$-th deployed scoring function is assigned failure probability $\delta_k = \delta/(2k^2)$. Since $\sum_{k=1}^{\infty} \frac{\delta}{2k^2} = \delta\frac{\pi^2}{12} < \delta$, a union bound over all deployed scoring functions ensures the overall failure probability is below $\delta$. Dividing by $N_t^{(o)}$ and identifying $\log(4k^2|\Lambda|/\delta) \leq 2\log(4k|\Lambda|/\delta) \leq 2\log(4U_t|\Lambda|/\delta)$, we obtain the desired bound. Under delayed deployment, $\widehat{g}^{(k)}$ is deployed only once $c_t N_t^{(o)} \geq 173\log(4/\delta_k)$, so the bound holds from deployment onward. $\square$

Proposition 3 establishes the desired safety guarantees for FPR control, ensuring that with probability at least $1 - \delta$, $\mathrm{FPR}(\widehat{g}_t, \widehat{\lambda}_t) \leq \alpha$ for all $t \geq t_0$ in the ASAT system. This comes with the trade-off, as $\psi_t(\delta)$ now contains the additional term $U_t$—the number of total updates—in the second logarithm term. In practice, however, $U_t$ is expected to be much smaller than $N_t^{(o)}$, as updates only occur periodically after collecting enough new OOD samples. As a result, the additional dependency on $U_t$ introduces only a minor increase in $\psi_t(\delta)$ compared to FSAT. This slight increase is outweighed by the advantage of using the learned $\widehat{g}_t$, which optimizes TPR. We emphasize that this guarantee applies to the calibration-safe variant described above. During the delayed deployment window for each new $\widehat{g}^{(k)}$, FPR control is maintained by the previous model $\widehat{g}^{(k-1)}$ with its already-valid threshold. Algorithm 1 and all experiments use the all-history estimator from Eq. 8 with a heuristic UCB; the empirical results demonstrate strong FPR control for this practical variant in Algorithm 1.

# D    Additional Experiments and Details

## D.1    Additional Baseline

For completeness, we compare ASAT against a naive binary classifier baseline. This baseline uses a two-layer neural network (with two output neurons) that matches the architecture and parameter size of ASAT, and is trained with cross-entropy loss on the ID and OOD data collected up to each time $t$. Since this classifier is not scoring-based, we do not use importance sampling. We update the classifier with the same frequency as ASAT, using the AdamW optimizer. Initially, the classifier is randomly initialized and predicts all samples as OOD until enough data is collected for its first update.

As shown in Figure 8, while the binary classifier can distinguish between ID and OOD points, it incurs significant FPR violations (exceeding 5%) because it does not explicitly optimize to maintain a low FPR while maximizing TPR. In contrast, ASAT adapts both thresholds and scoring functions to maximize TPR while ensuring strict FPR control. Hence, these results confirm that our ASAT framework successfully meets its dual objectives, offering a more reliable solution (via FPR control) for OOD detection.

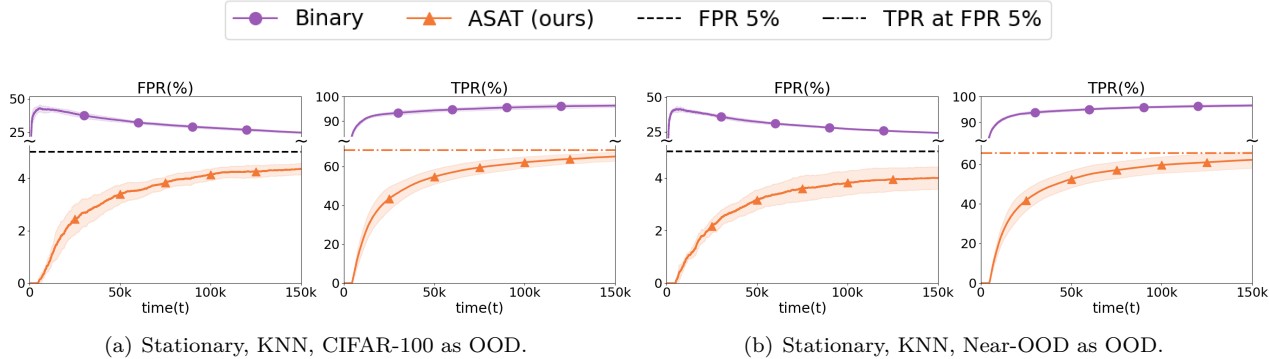

(a) Stationary, KNN, CIFAR-100 as OOD.  (b) Stationary, KNN, Near-OOD as OOD.

Figure 8: Comparison between ASAT and a naive binary classifier baseline. ASAT is initialized with KNN. CIFAR-10 as ID and CIFAR-100 and Near-OOD as OOD. Each method is repeated five times and is shown with mean and std. The binary classifier is trained using cross-entropy loss on the ID and OOD data collected up to time $t$, without importance sampling (since the classification is not based on score thresholding).

### D.2  Searching for Constants in Heuristic LIL UCB

We derived the following theoretical LIL-based upper confidence bound (UCB) to ensure FPR control for ASAT with time-varying scoring functions $\widehat{g}_t$ for $t \geq 1$:

$$\psi_t(\delta) := \sqrt{\frac{3c_t}{N_t^{(o)}}\left[2\log\log\left(\frac{3c_t N_t^{(o)}}{2}\right) + 2\log\left(\frac{4U_t|\Lambda|}{\delta}\right)\right]}, \tag{11}$$

where $U_t$ is the total number of scoring functions deployed in the system by time $t$. The inclusion of $U_t$ reflects the union bound required for FPR control given the time-varying scoring functions (see Appendix C). A similar UCB appears in the FSAT framework (Vishwakarma et al., 2024c), but without the $U_t$ term and with different definitions of $c_t$ and $N_t^{(o)}$. However, due to pessimistic constants in equation 6 (e.g., $|\Lambda|$ may be large for small $\eta$), we define a heuristic UCB:

$$\psi_t^H(\delta) := c_1 \sqrt{\frac{c_t}{N_t^{(o)}}\left[\log\log\left(c_2 c_t N_t^{(o)}\right) + \log\left(\frac{c_3}{\delta}\right)\right]},$$

with constants $c_1, c_2, c_3$. In FSAT, $c_1 = 0.5$, $c_2 = 0.75$, and $c_3 = 1.0$ were chosen via simulation of a $p$-biased coin to keep failure probability under 5%. In the ASAT main experiments, where we fix the optimization frequencies, we retain $c_2$ and $c_3$ but increase $c_1$ to 0.65 to account for the effect of $U_t$. The choice of $c_1$ depends on the number of scoring function updates, which requires adjustment if the optimization frequency changes.

### D.3  Choice of $\mathcal{G}$

Our ASAT framework is flexible in the choices of $\mathcal{G}$. In the main experiments, we use a class $\mathcal{G}_1$ of two-layer neural networks defined on top of the ResNet18 model (He et al., 2016). Let $z := r(x)$ be the output feature from the penultimate layer of the Resnet18, denoted as $r$, for an image $x$. A network $g \in \mathcal{G}_1$ takes $z$ as input and outputs a score $s \in \mathcal{S}$:

$$\mathcal{G}_1 := \{g : g(x) := \boldsymbol{W_2}\,\mathrm{ReLU}\left(\boldsymbol{W_1}r(x)\right)\},$$

where $\boldsymbol{W_1}, \boldsymbol{W_2}$ are learnable weight matrices of the two layers. The optimization procedures are detailed in the next section. This class of scoring functions leverages pre-extracted features for ID/OOD points and adapts the scoring functions *from scratch* to the specific OOD instances encountered by the system. This design is motivated by the observation that the backbone already performs heavy representation learning, and the 2-layer neural network as the scoring function only needs to learn a separation between ID and OOD samples, which makes the training computationally lightweight and scalable.

### D.4 Searching for Hyper-parameters

| Parameter | Values |
|---|---|
| optimizer | AdamW, SGD |
| learning rate for $g$ | 0.00001, 0.0001, 0.001, 0.01 |
| learning rate for $\lambda$ | 0.0001, 0.001, 0.01, 0.1 |
| weight decay rate for $g$ | 0.0001, 0.001, 0.01, 0.1, 0.0 |
| weight decay rate for $\lambda$ | 0.0001, 0.001, 0.01, 0.1, 0.0 |
| $\beta$ in (P2) | 0.5, 0.75, 1.0, 1.25, 1.5, 1.75, 2.0 |
| $\kappa$ for sigmoid smoothness | 1, 10, 50, 100 |

Table 3: Lists of hyper-parameters we swept over to find the best combination for training $g \in \mathcal{G}_1$ and $\lambda \in \Lambda$ in optimization (P2).

We determine the hyperparameters and their values for the optimization problem in (P2) over $\mathcal{G}_1$ via grid search. First, we find the best-performing hyperparameter combination for training $g \in \mathcal{G}_1$ in *offline* settings using i.i.d OOD and ID datasets. Then, we extend these choices to the ASAT settings, where only *dependent* OOD datasets are available. We verify that the same hyperparameters perform well in *online* settings, made possible by importance sampling techniques that preserve accurate FPR estimation in equation 2.

Table 3 lists the hyperparameters and their values we explored. Specifically, we run experiments on every different combination of these lists of hyperparameters. We use the AdamW optimizer (Kingma and Ba, 2014), with learning rates of 0.0001 for $g$ and 0.01 for $\lambda$, and a weight decay rate of 0.001 for L1 regularization for both $g$ and $\lambda$. We also set $\beta = 1.5$ and $\kappa = 50$.

### D.5 Sensitivity Analysis on Constants

We study the sensitivity of ASAT to the three problem-level constants $\alpha$, $p$, and $\gamma$ on synthetic data. For this analysis, we use ID and OOD scores drawn from $\mathcal{N}(\mu = 5.5, \sigma = 4)$ and $\mathcal{N}(\mu = -6, \sigma = 4)$ respectively, with a linear scoring function $g(x) = wx + b$. We run $T = 100k$ time steps with 5 seeds, varying one parameter at a time while fixing the others at baseline values we used in the main experiments ($\alpha = 0.05$, $p = 0.2$, $\gamma = 0.2$).

We sweep $\alpha \in \{0.01, 0.05, 0.1, 0.2\}$, $p \in \{0.05, 0.1, 0.2, 0.4\}$, and $\gamma \in \{0.05, 0.1, 0.2, 0.4\}$. Recall that $\alpha$ is the FPR tolerance set by the user, where smaller $\alpha$ enforces stricter safety at the cost of a more conservative threshold and lower best achievable TPR. The constant $p$ is the importance sampling probability, which controls the annotation budget, where larger $p$ means more human labels per time step. $\gamma$ is the OOD mixture rate in the data stream, which governs how frequently OOD samples arrive; ASAT does not use or require knowledge of $\gamma$, but it affects the rate at which OOD feedback accumulates and thus the speed of adaptation.

Overall, Table 4 demonstrates the robustness of ASAT across different system-level constants. Importantly, FPR is controlled below $\alpha$ in every setting, confirming that the safety guarantee is stable across these parameters. As expected, $\alpha$ controls the safety-performance trade-off, where smaller $\alpha$ requires a longer cold start and yields lower TPR, while population TPR converges toward the oracle TPR at each $\alpha$. The importance sampling probability $p$ has small effect on FPR/TPR (within ~1% across an 8× range); its primary effect is on annotation cost, which scales with $p$ (36k to 60k labels). The mixture ratio $\gamma$ affects cold-start duration (10k steps at $\gamma = 0.05$ vs. 1.3k at $\gamma = 0.40$), but all values converge to approximately the same FPR/TPR.

### D.6 Effects of Different Updating Frequencies

We define the optimization frequencies $\omega_i^{(ood)}$ as the number of newly human-labeled OOD points required since the last update for the $(i + 1)$th update of the scoring functions. In stationary settings, we fix $\omega_i^{(ood)}$ to be 100 for $i \in \{1, ..., 20\}$, 500 for $i \in \{21, 40\}$, and 1000 thereafter for consistency. This approach enables more frequent updates early on, with the frequency gradually decreasing as scoring functions improve. In nonstationary settings, we set $\omega_i^{(ood)}$ uniformly to avoid excessively infrequent updates after a shift.

**(a)** $\alpha$ sweep ($p = 0.2$, $\gamma = 0.2$). Last column is the best theoretically achievable TPR at each FPR-$\alpha$.

| $\alpha$ | Feasibility $t^*$ | Pop. FPR | Pop. TPR | Oracle TPR @ $\alpha$ |
|---|---|---|---|---|
| 0.01 | $66{,}004 \pm 366$ | $0.20 \pm 0.05\%$ | $49.64 \pm 2.00\%$ | $71.1\%$ |
| 0.05 | $2{,}534 \pm 102$ | $4.23 \pm 0.21\%$ | $87.41 \pm 0.63\%$ | $89.1\%$ |
| 0.10 | $672 \pm 27$ | $9.03 \pm 0.40\%$ | $93.76 \pm 0.25\%$ | $93.9\%$ |
| 0.20 | $209 \pm 24$ | $19.00 \pm 0.44\%$ | $97.71 \pm 0.06\%$ | $97.7\%$ |

**(b)** $p$ sweep ($\alpha = 0.05$, $\gamma = 0.2$). Last column is the number of total human annotations.

| $p$ | Feasibility $t^*$ | Pop. FPR | Pop. TPR | Total Human Labels |
|---|---|---|---|---|
| 0.05 | $2{,}534 \pm 102$ | $3.96 \pm 0.22\%$ | $86.75 \pm 0.66\%$ | $36{,}235 \pm 1{,}309$ |
| 0.10 | $2{,}534 \pm 102$ | $4.24 \pm 0.27\%$ | $87.43 \pm 0.60\%$ | $39{,}257 \pm 308$ |
| 0.20 | $2{,}534 \pm 102$ | $4.23 \pm 0.21\%$ | $87.41 \pm 0.63\%$ | $46{,}229 \pm 353$ |
| 0.40 | $2{,}534 \pm 102$ | $4.25 \pm 0.14\%$ | $87.47 \pm 0.46\%$ | $59{,}535 \pm 290$ |

**(c)** $\gamma$ sweep ($\alpha = 0.05$, $p = 0.2$).

| $\gamma$ | Feasibility $t^*$ | Pop. FPR | Pop. TPR |
|---|---|---|---|
| 0.05 | $10{,}084 \pm 543$ | $3.66 \pm 0.66\%$ | $85.73 \pm 1.89\%$ |
| 0.10 | $5{,}028 \pm 310$ | $3.88 \pm 0.42\%$ | $86.52 \pm 1.13\%$ |
| 0.20 | $2{,}534 \pm 102$ | $4.23 \pm 0.21\%$ | $87.41 \pm 0.63\%$ |
| 0.40 | $1{,}309 \pm 16$ | $4.41 \pm 0.20\%$ | $87.86 \pm 0.56\%$ |

Table 4: Sensitivity analysis on synthetic Gaussian data ($T = 100\text{k}$, 5 seeds). Each sweep varies one parameter while fixing the others at baseline ($\alpha = 0.05$, $p = 0.2$, $\gamma = 0.2$). The first column, *feasibility $t^*$*, is the first time step at which (Q1) becomes feasible and ASAT begins autonomous predictions; *Pop. FPR and TPR* are the population-level FPR and TPR at $T = 100\text{k}$ computed as in Eq. 7.

Additionally, $\omega_i^{(ood)}$ influences the width of LIL-based UCB. More frequent updates result in larger $U_t$, leading to a wider UCB. Thus, $\omega_i^{(ood)}$ presents a trade-off between faster OOD adaptation and larger UCB. More frequent updates of the scoring functions could also impact time efficiency, as each update requires additional computational resources.

To illustrate this, we run additional experiments in stationary settings with four different choices for $\omega_i^{(ood)}$: 100, 500, 1000, and 2500 uniformly for all $i \geq 1$. We fix $c_1 = 0.50$ for $\psi_t^H(\delta)$ across all frequencies, knowing this value is not large enough to provide strict FPR guarantees. This ensures a fair comparison across different frequencies. The results, shown in Figure 9, confirm our hypotheses: more frequent updates lead to faster adaptations but also result in a larger UCB width. More frequent updates impact time efficiency, with rough averages from five runs showing increased time costs. However, they also lead to better TPR convergence, which requires further exploration. Further research on update frequencies is left for future work.

### D.7 Additional Experiments with Different Scoring Functions

In the main experiments, we present results for EBO and KNN. Additionally, we run additional experiments using the following alternative scoring functions, with the same settings and constants as in the main experiments. Note that while these scoring functions remain fixed in both FSFT and FSAT, the scoring function is immediately updated to a new scoring function from $\mathcal{G}_1$ in ASAT. We use these scoring functions adapted by Yang et al. (2022).

1. **Energy Score (EBO).** EBO (Liu et al., 2020) uses an energy function derived from a discriminative model to distinguish OOD inputs from ID data. Energy scores are shown to align more with probability density and to be less prone to overconfidence than softmax scores.
2. **$K$-Nearest Neighbors (KNN).** KNN (Sun et al., 2022) uses non-parametric nearest-neighbors distance without strong assumptions about the feature space. In particular, KNN measures the distance between an input and its $k$ closest ID neighbors. We use $k = 50$ as in Open-OOD benchmarks.

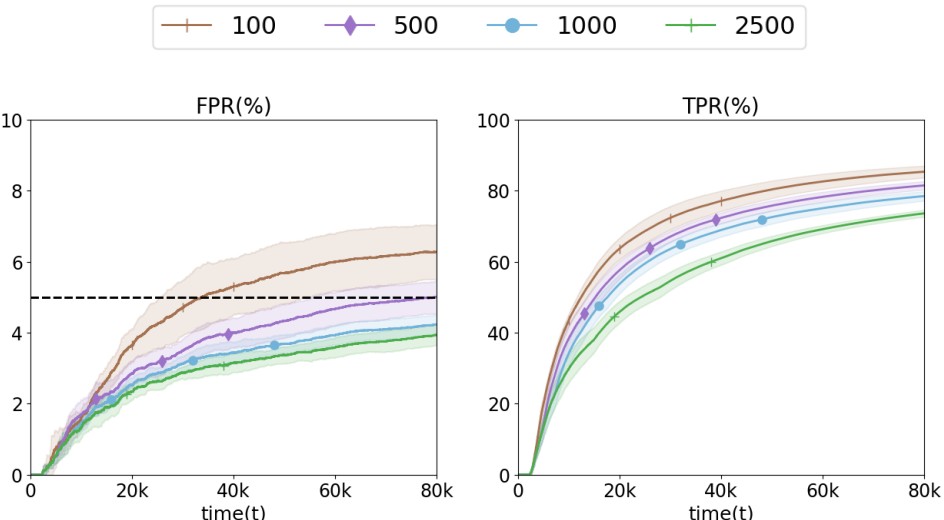

Figure 9: Results on the stationary setting with different update frequencies, with each method repeated 5 times (mean and std shown). CIFAR-10 as ID and Places365 as OOD. Optimization frequencies are set uniformly for 100, 500, 1000, and 2500 new OOD samples. We fix $c_1 = 0.5$ in the heuristic UCB $\psi_t^H(\delta)$.

3. **Mahalanobis Distance (MDS).** MDS (Lee et al., 2018b) computes the Mahalanobis distance between an input and the nearest class-conditional Gaussian distribution and outputs a confidence score based on Gaussian discriminant analysis (GDA).

4. **Virtual Logits Matching (VIM).** VIM (Wang et al., 2022) combines a class-agnostic score from the feature space with ID-dependent logits. It uses the residual of the feature against the principal space (we use dimension 10) and matches the logits with a constant scaling.

5. **OOD in Neural Networks (ODIN).** ODIN (Liang et al., 2018) uses softmax scores from DNNs, scales them with temperature, and applies gradient-based input perturbations to compute the score. We use a temperature of 1000 and a noise of 0.0014.

6. **Activation Shaping (ASH).** ASH (Djurisic et al., 2023) applies activation shaping, removing a large portion (e.g., 90%) of a sample's activation at a late layer, simplifying the rest. The method requires no extra training or significant model modifications.

7. **Simplified Hopfield Energy (SHE).** SHE (Zhang et al., 2023) computes a score based on Hopfield energy in a store-and-compare paradigm. It transforms penultimate layer features into stored patterns representing ID data, which serve as anchors for scoring new inputs.

Results for stationary and nonstationary OOD settings are shown in Figures 10 and 11, respectively.

## D.8 Additional Experiments on Different ID Datasets

In the main experiments, we use CIFAR-10 as the ID dataset across all experiments. Additionally, we present results for CIFAR-100 (Krizhevsky et al., 2009) as the ID dataset. For stationary settings, we consider two OOD datasets: (i) CIFAR-10 and (ii) Places365. In the nonstationary OOD settings, we evaluate the scenario where a shift occurs at $t = 50k$, transitioning from a Far-OOD mixture (MNIST, SVHN, and Texture) to a Near-OOD mixture (CIFAR-10, Tiny-ImageNet, and Places365). Results for stationary and distribution-shift settings are shown in Figures 12 and 13, respectively.

In addition, we have experiments in a large-scale setting using ViT-B/16 (Dosovitskiy et al., 2021) as the feature extractor and ImageNet-1k (Deng et al., 2009) as ID. In the stationary setting, we used OpenImage-O as OOD and compared ASAT against FSAT and FSFT with KNN and MDS scoring functions. As expected, ASAT outperformed FSAT in TPR while maintaining strict FPR control, which confirms that ASAT scales well to large datasets like ImageNet-1k and modern architectures like ViT. Results are shown in Figure 14.

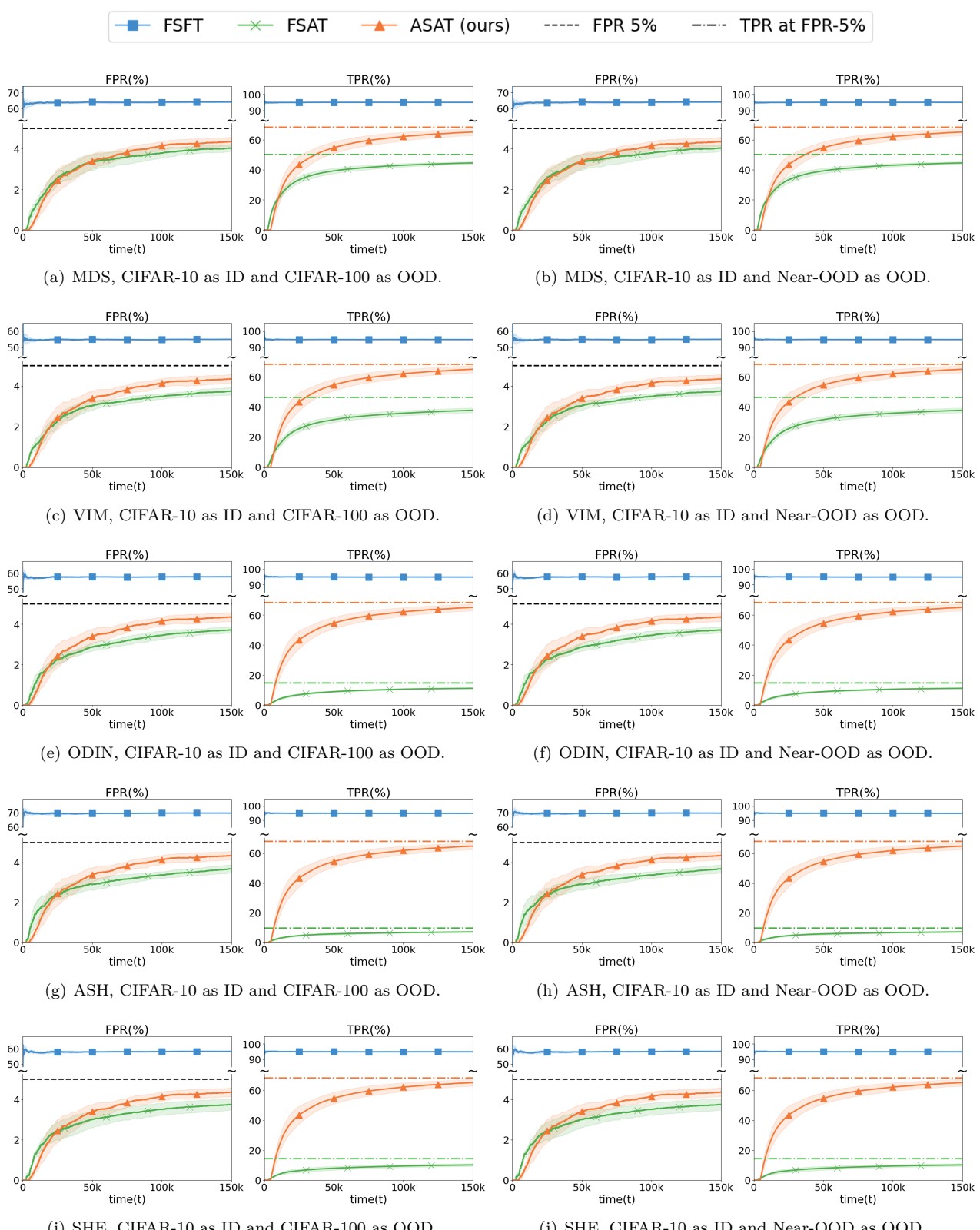

Figure 10: Results for stationary settings using CIFAR-10 as ID, with each method repeated 5 times (mean and std shown). Run with five additional scoring functions, MDS, VIM, ODIN, ASH, and SHE.

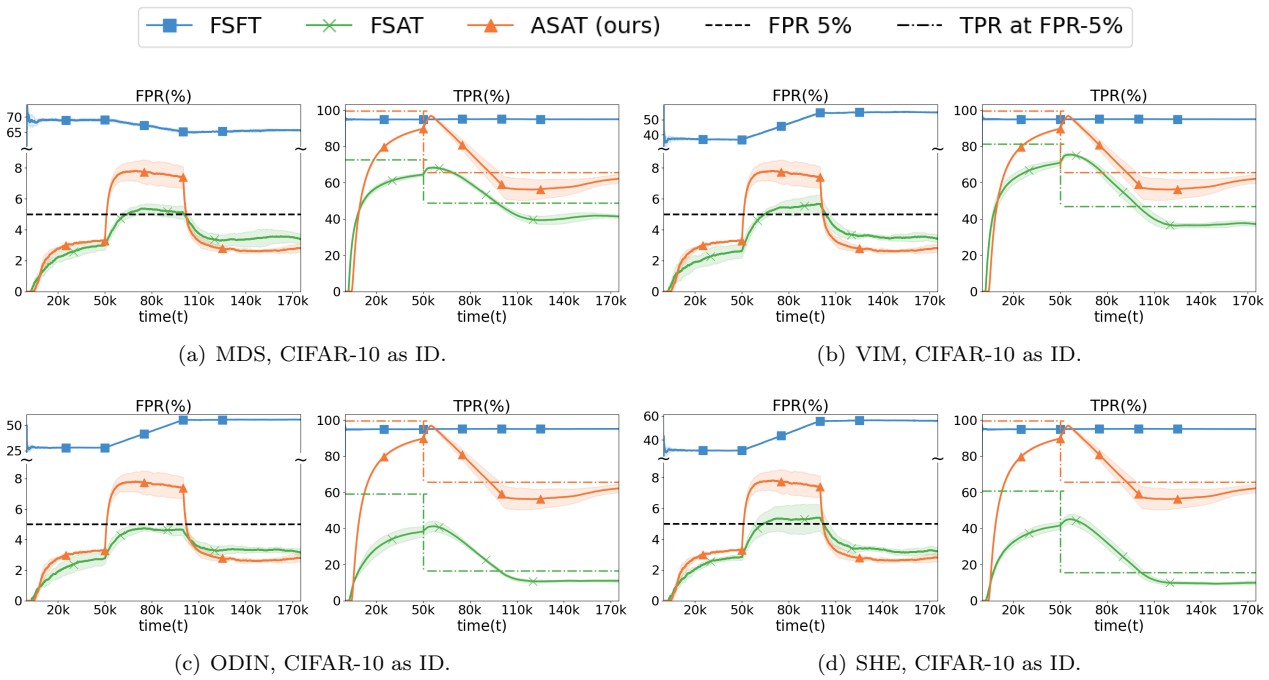

(a) MDS, CIFAR-10 as ID.

(b) VIM, CIFAR-10 as ID.

(c) ODIN, CIFAR-10 as ID.

(d) SHE, CIFAR-10 as ID.

Figure 11: Results for nonstationary OOD settings using CIFAR-10 as ID, with each method repeated 5 times (mean and std shown). OOD distribution shifts from Far-OOD to Near-OOD. Run with four additional scoring functions, MDS, VIM, ODIN, and SHE.

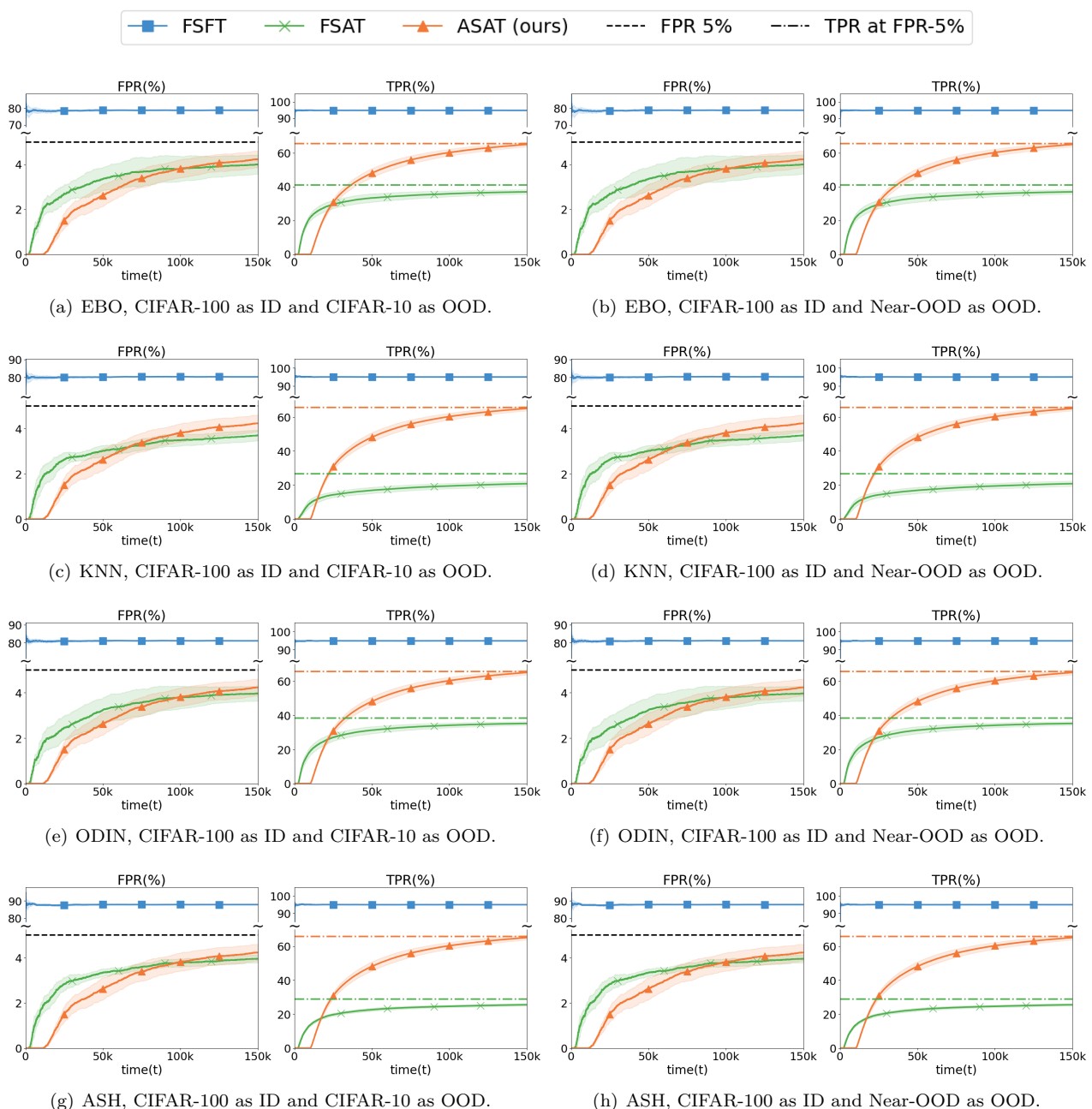

Figure 12: Results for stationary settings using CIFAR-100 as ID, with each method repeated 5 times (mean and std shown). Run with four scoring functions, EBO, KNN, ODIN, and ASH.

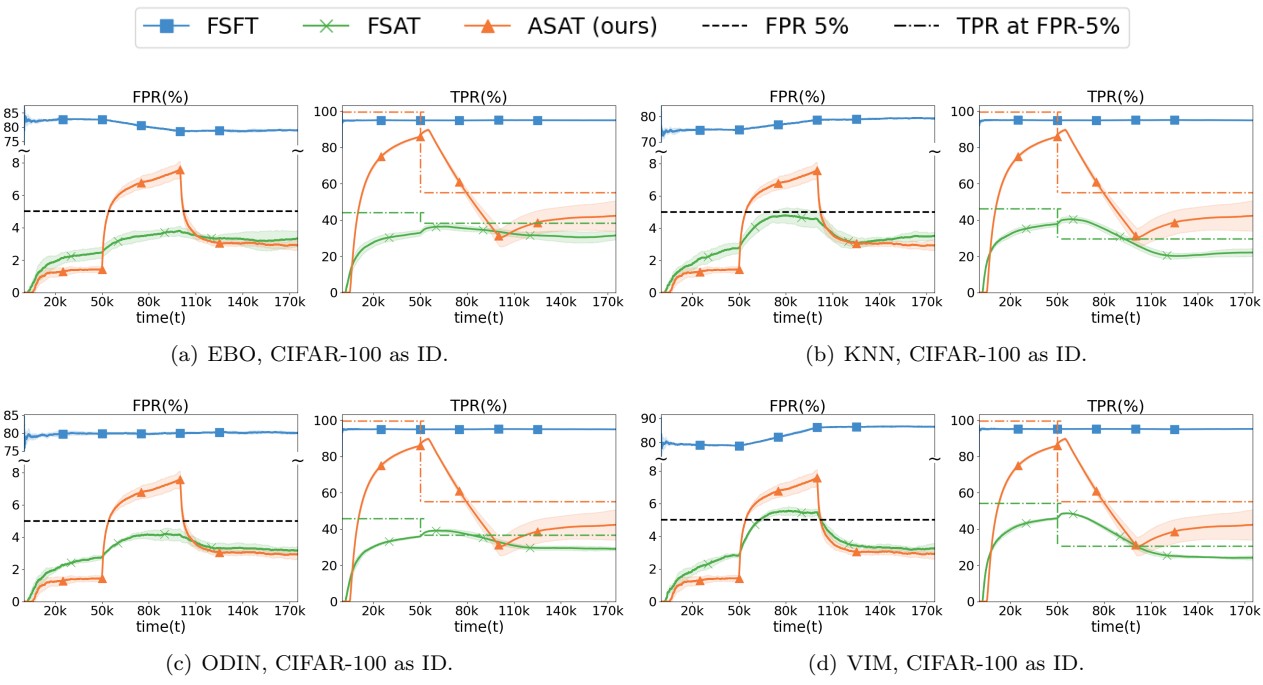

Figure 13: Results for nonstationary settings using CIFAR-100 as ID, with each method repeated 5 times (mean and std shown). OOD distribution shifts from Far-OOD to Near-OOD. Run with four additional scoring functions, EBO, KNN, ODIN, and VIM.

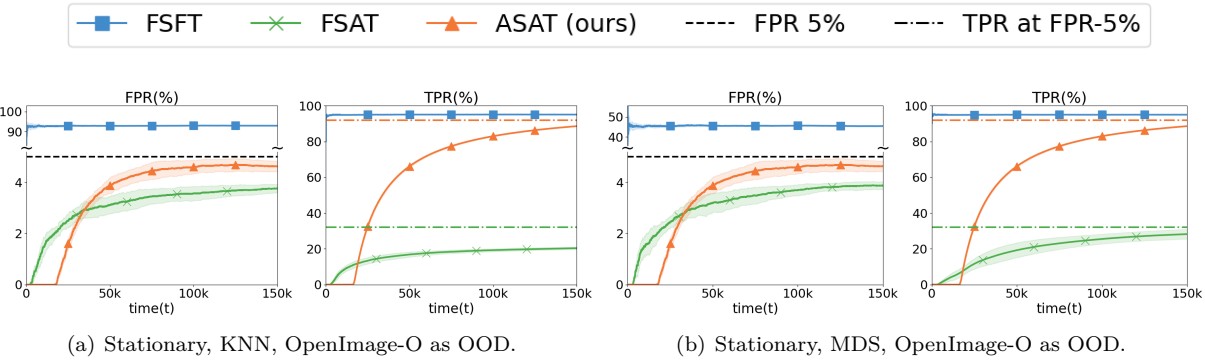

Figure 14: Results on the stationary setting for ImageNet-1k as ID where `ViT-B/16` is used as a feature extractor, with each method repeated 5 times (mean and std shown). KNN and MDS are the initial scoring functions and are fixed for FSFT and FSAT. The dotted-dash lines represent the *expected* TPR at FPR-5% for KNN, MDS, and $\widehat{g}^*$ (matching colors).

