# OpenReview forum: "ASAT: Adaptive Scoring and Thresholding with Human Feedback for Robust Out-of-Distribution Detection"
_TMLR — Accepted by TMLR_

### Review · Reviewer_xqfm · 2026-05-05

**Summary Of Contributions:**

**Summary**

This paper proposes ASAT, an adaptive human-in-the-loop framework for out-of-distribution (OOD) detection. The key idea is to use human feedback collected during deployment to update both the scoring function and the decision threshold, with the goal of maximizing TPR while controlling FPR.

**Strength**

The paper addresses an important deployment-oriented problem, formulates OOD detection under an explicit FPR constraint, and provides a practically motivated human-feedback mechanism. The empirical evaluation is reasonably broad: the paper considers stationary and nonstationary OOD streams, multiple initial scoring functions, several ID/OOD dataset combinations, and additional experiments on larger-scale ImageNet-1k settings.

**Weakness**

The theoretical guarantee for strict FPR control under adaptive scoring appears incomplete. The martingale argument in Lemma 2 and Proposition 2 is established for a fixed scoring function $g$, whereas ASAT learns $\hat g_t$ from the same OOD feedback samples that are later reused for FPR estimation, threshold calibration, and threshold selection in (Q1). This adaptive reuse of data may make $\widehat{\mathrm{FPR}}_t(\hat g_t,\lambda)$ overly optimistic relative to the true $\mathrm{FPR}(\hat g_t,\lambda)$, unless additional arguments are provided to control this dependence. As a result, the strongest claim of strict FPR control for the full adaptive method is not fully justified by the current proof.

**Audience:**

Yes

**Audience Explanation:**

The paper studies an timely problem: how to adapt OOD detection systems during deployment while explicitly controlling false positives. This is relevant to researchers working on OOD detection, selective prediction, human-in-the-loop learning, online calibration, and safety-critical ML deployment. The idea of combining adaptive scoring with adaptive thresholding under human feedback is practically motivated and could stimulate further work on deployable OOD detection systems.

**Broader Impact Concerns:**

The paper should include a broader impact or limitations discussion emphasizing that the guarantee depends on the validity of the stationarity and calibration assumptions, that human feedback is assumed accurate, and that additional safeguards.

**Claims And Evidence:**

Yes

**Claims Explanation:**

Yes, with an important caveat. The empirical claims are generally supported by clear and reasonably extensive evidence. The paper evaluates ASAT in both stationary and nonstationary OOD settings, compares it against FSFT and FSAT, and includes additional experiments across multiple initial scoring functions, ID/OOD dataset choices, and a larger-scale ImageNet-1k setting. These results provide convincing empirical evidence that adapting the scoring function using human-labeled OOD feedback can improve TPR over threshold-only adaptation while maintaining low empirical FPR in many stationary settings.

However, the strongest theoretical claim is not fully justified by the current proof. Lemma 2 and Proposition 2 establish the martingale argument for a fixed scoring function $g$, whereas ASAT learns $\hat g_t$ in (P2) from the same OOD feedback samples that are later reused in (Q1) for FPR estimation and threshold calibration. This adaptive reuse of data could make $\text{FPR}_t(\hat g_t,\lambda)$ overly optimistic relative to the true $\text{FPR}(\hat g_t,\lambda)$, unless additional arguments are provided to control this dependence.

Thus, I consider the empirical evidence to be largely convincing, but the theoretical guarantee should either be repaired.

**Requested Changes:**

1. The proof attempts to extend the fixed-score FSAT guarantee to time-varying deployed scoring functions via a union bound over updates. However, it remains unclear whether this fully accounts for the fact that ASAT trains $\hat g_t$ using the same OOD feedback samples that are later reused for FPR estimation and threshold calibration in (Q1). The authors should clarify the adaptivity argument, or consider sample splitting between OOD feedback used for learning $\hat g_t$ in (P2) and samples used for FPR calibration.

2. The experiments use the heuristic UCB $\psi_t^H$ rather than the theoretical bound $\psi_t$, with constants such as $c_1$ tuned for ASAT. The authors should report results using the theoretical UCB, or clearly state that the empirical control relies on a heuristic bound. A brief sensitivity analysis over $c_1$ would also help assess whether the FPR control is robust or depends on constant tuning.

3. The claim of **strict FPR control at all times** should be tempered, the theoretical guarantee holds for stationary settings. The authors should clarify that rigorous FPR control is valid under stationary assumptions.

---

> ### Author Response · Authors · 2026-05-20
>
> Thank you for the constructive and technically important feedback.
>
> ---
>
> > The theoretical guarantee for strict FPR control under adaptive scoring appears incomplete. The martingale argument in Lemma 2 and Proposition 2 is established for a fixed scoring function, whereas ASAT learns $\hat{g}$ from the same OOD feedback samples that are later reused for FPR estimation, threshold calibration, and threshold selection in (Q1). This adaptive reuse of data may make $\widehat{\mathrm{FPR}}$ overly optimistic relative to the true $\mathrm{FPR}$, unless additional arguments are provided to control this dependence. As a result, the strongest claim of strict FPR control for the full adaptive method is not fully justified by the current proof.
> >
> > The proof attempts to extend the fixed-score FSAT guarantee to time-varying deployed scoring functions via a union bound over updates. However, it remains unclear whether this fully accounts for the fact that ASAT trains $\hat{g}$ using the same OOD feedback samples that are later reused for FPR estimation and threshold calibration in (Q1). The authors should clarify the adaptivity argument, or consider sample splitting between OOD feedback used for learning $\hat{g}$ in (P2) and samples used for FPR calibration.
>
> Thank you for pointing this out. We agree that the current proof does not fully justify the strongest version of the adaptive FPR-control claim as you point out, which we clarify and address below.
>
> The core issue is that the FPR estimator in Eq. (2) sums over all $u = 1, \ldots, t$, including pre-deployment terms ($u \leq \tau_k$) where $\tau_{k}$ denotes the training time of $\hat{g}^{(k)}$ (i.e., the $k$-th trained scoring function in the system). Critically, the scoring function $\hat{g}^{(k)}$ may depend on $x_u$ through training. For these terms, two problems arise: (i) $\hat{g}^{(k)}$ is not $\mathcal{F}\_{u}$-measurable (it depends on future data $x_{u+1}, \ldots, x_{\tau_k}$), breaking adaptedness in Lemma 2, and (ii) $\hat{g}^{(k)}$ and $x_u$ are dependent, breaking the conditional mean-zero property that Lemma 2 requires via Lemma 1. Importantly, this does not affect terms at $u > \tau_k$, where $\hat{g}^{(k)}$ is fully determined and $x_u$ is fresh. To address this, we revise the formal guarantee to use temporal splitting and a post-training FPR estimator for each $\hat{g}^{(k)}$ so that *calibration uses only the post-training samples*:
>
> $$\widehat{\mathrm{FPR}}^{\mathrm{post}}\_{k,t}(\hat{g}^{(k)}, \lambda) := \frac{\sum_{u=\tau_k+1}^{t} Z_u \cdot \mathbb{1}\{(\hat{g}^{(k)}(x_u) > \lambda\})}{\sum_{u=\tau_k+1}^{t} Z_u}$$
>
> Since every term has $u > \tau_k$, $\hat{g}^{(k)}$ is $\mathcal{F}\_{\tau_k} \subseteq \mathcal{F}\_{u-1}$-measurable and $x_u$ is fresh. The martingale argument applies, and the proof structure of Proposition 1 carries through with post-training quantities. To avoid a cold-start ($\lambda = +\infty$) after each scoring function update, we can use delayed deployment. A candidate $\hat{g}^{(k)}$ is trained at time $\tau_k$ but the previous $\hat{g}^{(k-1)}$ remains active, while $\hat{g}^{(k)}$ is calibrated in the background using only samples collected after $\tau_k$. Once (Q1) becomes feasible for $\hat{g}^{(k)}$, we deploy it with its calibrated threshold. Since all calibration samples for $\hat{g}^{(k)}$ occur after $\hat{g}^{(k)}$ is fixed, the same martingale argument applies. During the waiting period, FPR control is maintained by the already calibrated active model. The cost is a delay in deploying the improved scoring function.
>
> We have updated the main text including Proposition 1 and the proofs in Appendix C in detail to reflect the post-training estimator and delayed deployment in the revision. We note that our experiments use the all-history estimator (Eq. 2) with a heuristic UCB, which should be interpreted as empirical evidence for feasibility and performance rather than as directly covered by the revised theorem. It would be an interesting future direction to understand the version without sample splitting which experimentally works well, as remarked in the revised Appendix C (see Remark 1).

---

> ### Author Response · Authors · 2026-05-20
>
> > The experiments use the heuristic UCB $\psi_t^H$ rather than the theoretical bound $\psi_t$, with constants such as $c_1$ tuned for ASAT. The authors should report results using the theoretical UCB, or clearly state that the empirical control relies on a heuristic bound. A brief sensitivity analysis over $c_1$ would also help assess whether the FPR control is robust or depends on constant tuning.
>
> We agree that the distinction between the theoretical bound $\psi_t(\delta)$ and the heuristic bound $\psi_t^H(\delta)$ should be clearer. The experiments in the current submission use the heuristic bound. Regarding sensitivity to $c_1$, Appendix D.6 provides partial evidence, as it explores different update frequencies at fixed $c_1 = 0.5$.  We have revised the draft to state explicitly in the main text that the empirical curves are produced using the heuristic UCB rather than theoretical ones, and that robustness to the heuristic constant is an empirical consideration not covered by the formal theorem.
>
> > The claim of strict FPR control at all times should be tempered, the theoretical guarantee holds for stationary settings. The authors should clarify that rigorous FPR control is valid under stationary assumptions.
>
> Thank you for the suggestion. We have revised the abstract and the main text to state that the formal guarantee holds under stationary assumptions. For nonstationary OOD streams, we also made explicit that brief FPR violations may occur after distribution shifts, as already observed in the experiments (Figures 5 and 6), and that the method should be understood as adapting after such shifts rather than guaranteeing uniform control through them.
>
> > The paper should include a broader impact or limitations discussion emphasizing that the guarantee depends on the validity of the stationarity and calibration assumptions, that human feedback is assumed accurate, and that additional safeguards.
>
> We have expanded our current Section 8 (Limitations and Future Work) to emphasize these points as suggested. Thank you for the suggestion.
>
> ----------
>
> We hope these clarifications and the revised theory address the concerns. If any specific points remain, we are happy to discuss further.

---

### Review · Reviewer_S6RE · 2026-05-07

**Summary Of Contributions:**

**Summary of Contributions**:

The main contribution of this paper is the ASAT framework, which extends conventional OOD detection by adapting both the scoring function and the decision threshold during deployment. Unlike fixed-threshold methods that target high TPR but leave FPR uncontrolled, ASAT leverages human feedback to collect real-world OOD samples and update the detector online. The paper further provides theoretical guarantees for FPR control under stationary conditions and demonstrates through OpenOOD experiments that ASAT improves TPR over fixed-scoring and fixed-threshold baselines while keeping FPR below the desired tolerance.

**Strengths**:
1. Human-in-the-loop design is meaningful in OOD because incorporating human feedback addresses the scarcity of OOD data during training and allows adaptation to real-world scenarios.
2. It's natural and novel to jointly adapt scoring functions and thresholds, extending prior approaches that only used fixed scoring or fixed scoring with adaptive thresholds.
3. Provides a theoretical analysis for FPR control under stationary OOD and extensive experiments on OpenOOD.

**Weaknesses**:
1. The method depends on human feedback and assumes it is noise-free, but in practice, inaccurate labels are inevitable and could degrade performance.
2. Low TPR and high thresholds at early stages require more human annotations, increasing cost and reducing efficiency during initial deployment. It also conflicts with the target to minimize human intervention. Clarification on this is needed.
3. The framework involves multiple hyperparameters, which may require careful tuning to achieve optimal performance in practice. Moreover, the paper appears to tune hyperparameters using offline data, which should be clarified because it conflicts with the claimed deployment setting where OOD data is unavailable. Sensitivity analyses and annotation cost analysis for p, gamma, alpha, beta, and kappa are needed.

**Audience:**

Yes

**Audience Explanation:**

OOD detection is critical in safety-sensitive domains like medical imaging and autonomous systems. This paper provides an online, adaptive framework with theoretical guarantees and empirical validation. Lowering false positives reduces unnecessary interventions and costs, while adaptive updates allow the system to handle evolving OOD inputs more efficiently. This makes the work relevant to researchers in OOD detection and reliable ML for safety-critical applications.

**Broader Impact Concerns:**

No broader impact concerns.

**Claims And Evidence:**

Yes

**Claims Explanation:**

- Theoretically, the claims are supported by Proposition 1 and detailed derivations showing FPR control with high probability under stationary OOD conditions.
- Empirically, Experiments on OpenOOD benchmarks, including CIFAR-10 ID and various OOD datasets, consistently show ASAT achieves higher TPR while controlling FPR compared to baselines.

**Requested Changes:**

1. Clarify the human-labeling cost, especially during the cold-start phase.
2. Clarify how hyperparameters are selected in realistic deployment settings where OOD data is limited beforehand, since the current experiments use grid-searched hyperparameters.
3. Provide sensitivity analyses for p, gamma, alpha, beta, and kappa, as these parameters directly affect FPR control, adaptation speed, and annotation cost.
4. Make the threshold convention consistent throughout the paper, since the paper alternates between g>lambda and g>=lambda
5. Correct the mismatched scoring-function names in the figure captions and panels in Figures 12 and 13.

---

> ### Author Response · Authors · 2026-05-20
>
> Thank you for the constructive feedback.
>
> ---
>
> > The method depends on human feedback and assumes it is noise-free, but in practice, inaccurate labels are inevitable and could degrade performance.
>
> We clarify that our setup assumes *expert feedback* as described in Section 1, where the system flags OOD inputs to receive "expert human intervention." In domains like medical diagnosis, such expert labels can be reasonably assumed to be accurate. That said, we agree that there can be settings with label noise. There is rich literature in crowdsourcing on label noise mitigation. We mention this in Section 8, where we outline how techniques for learning with label noise, such as unbiased estimators from Natarajan et al. (2013), could potentially be incorporated into the FPR estimation step. We view robustness to noisy human feedback as an important extension, but separate from the main focus of this work: jointly adapting scoring functions and thresholds under the FPR-control setting.
>
> > Low TPR and high thresholds at early stages require more human annotations, increasing cost and reducing efficiency during initial deployment. It also conflicts with the target to minimize human intervention. Clarification on this is needed. Clarify the human-labeling cost, especially during the cold-start phase.
>
> We view the cold-start phase as a *deliberate safety mechanism*. The system initially sets the threshold $\lambda = +\infty$, so every sample gets a human label until enough OOD feedback has been collected for the confidence bound $\psi_t(\delta)$ to satisfy the FPR constraint. This is necessary to maintain the FPR control early in deployment, and it does not conflict with the goal of minimizing human intervention in the long run. Rather, it reflects the trade-off between safety and efficiency. As $t$ grows, $\psi_t(\delta)$ shrinks, the threshold becomes less conservative, TPR increases, and fewer samples require human labels. As discussed in Section 5.2, ASAT can initially have a longer cold start than FSAT because adapting both scoring functions and thresholds requires a wider confidence bound and more OOD samples. However, this early cost is offset in our experiments by **ASAT's ability to achieve substantially higher TPR later in deployment than FSAT**.

---

> ### Author Response · Authors · 2026-05-20
>
> > The framework involves multiple hyperparameters, which may require careful tuning to achieve optimal performance in practice. Moreover, the paper appears to tune hyperparameters using offline data, which should be clarified because it conflicts with the claimed deployment setting where OOD data is unavailable. Sensitivity analyses and annotation cost analysis for p, gamma, alpha, beta, and kappa are needed. Clarify how hyperparameters are selected in realistic deployment settings where OOD data is limited beforehand, since the current experiments use grid-searched hyperparameters.
>
> Thank you for raising this. We clarify the roles of these quantities and add sensitivity analysis. We included this analysis in the revised Appendix D.5, Table 4.
>
> - **$\alpha$** is the target FPR tolerance set by the practitioner, which is not a tuned hyperparameter. Smaller $\alpha$ gives more conservative thresholds and lower TPR.
> - **$p$** is the importance sampling probability chosen based on annotation budget, which is also not tuned. Lower $p$ reduces labeling among predicted-ID samples but increases variance and slows adaptation.
> - **$\gamma$** is the OOD mixture rate in the data stream and simulations, which is not a parameter of the algorithm. ASAT does not require knowing $\gamma$.
> - **$\beta$** and **$\kappa$** are optimization hyperparameters in (P2). They affect the quality of the learned scoring function, and therefore TPR/adaptation speed. Under the formal threshold-calibration step, FPR control is enforced separately by (Q1).
>
> To address the request, we simulate ASAT on synthetic data with ID scores drawn from $\mathcal{N}(\mu=5.5, \sigma=4)$ and OOD scores from $\mathcal{N}(\mu=-6, \sigma=4)$. We run $T=100$k time steps with 5 seeds, varying one parameter at a time while fixing the others at baseline ($\alpha=0.05, p=0.2, \gamma=0.2$). We use the same ASAT thresholding and sampling logic in this synthetic score-level setting. Feasibility $t^*$ is the first time step at which (Q1) becomes feasible and ASAT begins autonomous predictions; Pop. FPR and TPR are the population-level FPR and TPR at $T=100$k.
>
> **$\alpha$ sweep** ($p=0.2, \gamma=0.2$):
> | $\alpha$ | Feasibility $t^*$ | Pop. FPR | Pop. TPR | Oracle TPR @ $\alpha$ |
> |---|---|---|---|---|
> | 0.01 | 66,004 ± 366 | 0.20 ± 0.05% | 49.64 ± 2.00% | 71.1% |
> | 0.05 | 2,534 ± 102 | 4.23 ± 0.21% | 87.41 ± 0.63% | 89.1% |
> | 0.10 | 672 ± 27 | 9.03 ± 0.40% | 93.76 ± 0.25% | 93.9% |
> | 0.20 | 209 ± 24 | 19.00 ± 0.44% | 97.71 ± 0.06% | 97.7% |
>
> **$p$ sweep** ($\alpha=0.05, \gamma=0.2$):
> | $p$ | Feasibility $t^*$ | Pop. FPR | Pop. TPR | Total Human Labels |
> |---|---|---|---|---|
> | 0.05 | 2,534 ± 102 | 3.96 ± 0.22% | 86.75 ± 0.66% | 36,235 ± 1,309 |
> | 0.10 | 2,534 ± 102 | 4.24 ± 0.27% | 87.43 ± 0.60% | 39,257 ± 308 |
> | 0.20 | 2,534 ± 102 | 4.23 ± 0.21% | 87.41 ± 0.63% | 46,229 ± 353 |
> | 0.40 | 2,534 ± 102 | 4.25 ± 0.14% | 87.47 ± 0.46% | 59,535 ± 290 |
>
> **$\gamma$ sweep** ($\alpha=0.05, p=0.2$):
> | $\gamma$ | Feasibility $t^*$ | Pop. FPR | Pop. TPR |
> |---|---|---|---|
> | 0.05 | 10,084 ± 543 | 3.66 ± 0.66% | 85.73 ± 1.89% |
> | 0.10 | 5,028 ± 310 | 3.88 ± 0.42% | 86.52 ± 1.13% |
> | 0.20 | 2,534 ± 102 | 4.23 ± 0.21% | 87.41 ± 0.63% |
> | 0.40 | 1,309 ± 16 | 4.41 ± 0.20% | 87.86 ± 0.56% |
>
> Four observations. (1) *FPR is controlled below $\alpha$ in every setting* in this synthetic study, suggesting that empirical FPR control is stable across these parameters. (2) *$\alpha$ controls the safety-performance trade-off as intended*: smaller $\alpha$ requires a longer cold start and yields lower TPR (due to the more conservative threshold), while population TPR converges toward the oracle TPR at each $\alpha$. (3) $p$ has small effect on FPR/TPR (within ~1% across an 8× range); its primary effect is on annotation cost and convergence, which scales with $p$ (36k to 60k labels). (4) $\gamma$ affects cold-start duration (10k steps at $\gamma=0.05$ vs. 1.3k at $\gamma=0.40$), but all values move toward the same FPR/TPR. Together, these results are consistent with the expected behavior of ASAT.
>
> We also clarify the hyperparameter selection. In the experiments, some optimization hyperparameters, including $\beta$ and $\kappa$, were selected using a small held-out subset of benchmark OOD data. This was part of the experimental protocol, not an assumption required by ASAT at deployment. Since these hyperparameters affect scoring-function updates, in deployment they *can be selected after ASAT collects an initial batch of human-labeled OOD feedback*. We will clarify this distinction explicitly in the revision.
>
> > Make the threshold convention consistent throughout the paper, since the paper alternates between $g > \lambda$ and $g \geq \lambda$. Correct the mismatched scoring-function names in the figure captions and panels in Figures 12 and 13.
>
> Thank you for catching these. We revised the draft to consistently use $g(x) > \lambda$ throughout and corrected the figure captions in the revision.

---

### Review · Reviewer_wC8m · 2026-05-11

**Summary Of Contributions:**

This paper studies out-of-distribution detection, where an algorithm is required to classify whether a given data are inliers or outliers. The goal is to maximize the true positive rate while maintaining the false positive rate below a desired level. The authors propose a new algorithm that adaptively changes the scoring function and detection threshold based on input data and human feedback.

Strengths:
1. Adaptively changing the outlier detection algorithm is important in many real-world applications. The authors design an interesting algorithm that can adapt the scoring function and threshold to changing input data.
2. Experiment results demonstrate the effectiveness of the proposed algorithm against fixed threshold algorithms and naive binary classification.

Weakeness:
Some important details in the theoretical analysis are not presented clearly, particularly the input distribution assumptions. See the requested changes for more details.

**Audience:**

Yes

**Audience Explanation:**

Out-of-distribution detection is an important topic in machine learning. The paper should be interesting

**Broader Impact Concerns:**

There are no obvious ethical concerns regarding the topic and the experiments.

**Claims And Evidence:**

Yes

**Claims Explanation:**

The authors give theoretical guarantees that FPR can be controlled below a certain threshold. Experiment results demonstrate the effectiveness of the proposed algorithm against fixed threshold algorithms and naive binary classification.

**Requested Changes:**

Critical changes:

I hope the authors could clarify some aspects of the theoretical result (Proposition 1). My intuition is that the performance of the algorithm should somehow depend on the inlier and outlier distributions: in the extreme case when they are the same, it would be impossible to perform OOD, while when they are very far apart (e.g., total variation distance is 1), it should be fairly easy. The theorem statement and the performance guarantee should rely on some distance metric between the inlier and outlier distributions, but I do not see it in the theoretical results. It would be great if the authors could clarify where does the discrepancy between inliers and outliers come into play.


Changes that would strengthen the paper:

The authors set the false positive threshold at 0.05, but the desired False Positive Rate may be different for various scenarios. The authors should conduct experiments on one or two different FPR thresholds to demonstrate the generality of the proposed algorithm. This would strengthen the paper.

---

> ### Author Response · Authors · 2026-05-20
>
> Thank you for the constructive feedback and thoughtful questions.
>
> ---
>
> > I hope the authors could clarify some aspects of the theoretical result (Proposition 1). My intuition is that the performance of the algorithm should somehow depend on the inlier and outlier distributions: in the extreme case when they are the same, it would be impossible to perform OOD, while when they are very far apart (e.g., total variation distance is 1), it should be fairly easy. The theorem statement and the performance guarantee should rely on some distance metric between the inlier and outlier distributions, but I do not see it in the theoretical results. It would be great if the authors could clarify where does the discrepancy between inliers and outliers come into play.
>
> Thank you for this question. ASAT's core motivation is safe deployment of OOD systems in critical domains, where we want to strictly maintain FPR below a user's threshold $\alpha$ while maximizing TPR. Here, the key distinction is that Proposition 1 is a **safety guarantee** (FPR $\leq \alpha$). Importantly, *the FPR control by definition is with respect to the OOD distribution $\mathcal{D}_0$* and does not require a separation condition between $\mathcal{D}_0$ and $\mathcal{D}_1$.
>
> In particular, the distributional separation enters implicitly through the *achievable TPR*. As discussed in Section 2.1 (Population-level FPR and TPR), for a fixed scoring function $g$, the trade-off between FPR and TPR is governed by the overlap of the score distributions $g(x)$ under $\mathcal{D}_0$ and $\mathcal{D}_1$ (see Figure 3). When $\mathcal{D}_0$ and $\mathcal{D}_1$ induce significantly overlapping score distributions, the TPR achievable under FPR $\leq \alpha$ is low; in extreme cases where they induce the same score distributions, the best TPR is indeed at most $\alpha$ up to discretization effects. When the distributions are well-separated (e.g., TV distance close to 1), the optimal TPR approaches 1. This is reflected in our experiments through the oracle TPR at FPR = $\alpha$ (e.g., Table in our response to Reviewer S6RE), which represents the best achievable TPR for a given distributional separation.
>
> Therefore, Proposition 1 guarantees that ASAT controls the FPR budget $\alpha$, regardless of the distributions. The level of TPR achievable (at FPR level $\alpha$) depends on the separation between $\mathcal{D}_0$ and $\mathcal{D}_1$, which manifests in the score-distribution overlap. Importantly, ASAT is **designed to improve this separation over time by updating the scoring function, so it is not limited to the TPR achievable by the initial fixed score**. The achievable TPR is still distribution-dependent, but adaptive scoring allows ASAT to approach a better operating point when the collected OOD feedback contains useful separation information. We added a remark after Proposition 1 making this distinction explicit.
>
> > The authors set the false positive threshold at 0.05, but the desired False Positive Rate may be different for various scenarios. The authors should conduct experiments on one or two different FPR thresholds to demonstrate the generality of the proposed algorithm.
>
> Thank you for the suggestion. In our response to Reviewer S6RE, we provide the analysis on the FPR tolerance $\alpha$ in detail. We sweep $\alpha \in \{0.01, 0.05, 0.10, 0.20\}$ on synthetic data ($T=100$k, 5 seeds) and observe that FPR is controlled below $\alpha$ in every setting, with population TPR moving toward the best achievable TPR at each $\alpha$. We reproduce the table here for convenience:
>
> | $\alpha$ | Feasibility $t^*$ | Pop. FPR | Pop. TPR | Oracle TPR @ $\alpha$ |
> |---|---|---|---|---|
> | 0.01 | 66,004 ± 366 | 0.20 ± 0.05% | 49.64 ± 2.00% | 71.1% |
> | 0.05 | 2,534 ± 102 | 4.23 ± 0.21% | 87.41 ± 0.63% | 89.1% |
> | 0.10 | 672 ± 27 | 9.03 ± 0.40% | 93.76 ± 0.25% | 93.9% |
> | 0.20 | 209 ± 24 | 19.00 ± 0.44% | 97.71 ± 0.06% | 97.7% |
>
> These results provide additional evidence that ASAT behaves as expected across different FPR targets. We included this analysis in the revised Appendix D.5.
>
> ---
>
> We hope these clarifications address the questions. If any specific points remain, we are happy to discuss further.

---

### Decision · Action_Editor_hxRR · 2026-06-20

**Recommendation:** Accept as is

**Audience:**

Yes

**Audience Explanation:**

Out-of-distribution (OOD) detection is a core research topic in trustworthy machine learning (ML), with wide relevance to safety-critical deployment scenarios. This work proposes a practical human-in-the-loop framework that jointly adapts both scoring functions and decision thresholds during deployment. The findings are highly relevant to researchers in OOD detection, human-in-the-loop learning, calibration, and trustworthy ML deployment. The deployment-based design and theoretical-empirical combined contributions match the scope of TMLR and will be of interest to the journal's audience.

**Claims And Evidence:**

Yes

**Claims Explanation:**

The claims of the submission are well supported by both theoretical and empirical evidence. Theoretically, the authors provide guarantees for FPR control under some conditions. Empirically, the authors conduct extensive experiments on standard benchmarks. Additional sensitivity analyses on FPR tolerance and hyperparameters validate the robustness of the method. During the rebuttal period, the authors have thoroughly responded to all reviewers' concerns, supplemented missing analyses, and revised the manuscript accordingly, with all core questions resolved satisfactorily.